# Temporal Changes in Flow Regime along the River Vistula

Ewa Bogdanowicz *, Emilia Karamuz and Renata Julita Romanowicz

Department of Hydrology and Hydrodynamics, Institute of Geophysics Polish Academy of Sciences,
Ks. Janusza 64, 01-542 Warsaw, Poland; emilia_karamuz@igf.edu.pl (E.K.); romanowicz@igf.edu.pl (R.J.R.)
* Correspondence: ewabgd@igf.edu.pl

**Abstract:** The flow regime in the River Vistula is influenced by climatic and geographical factors and human intervention. In this study, we focus on an analysis of flow and precipitation variability over time and space following the course of the River Vistula. Multi-purpose statistical analyses of a number of runoff and precipitation characteristics were performed to present a general overview of the temporal and spatial changes. Since the important feature of the hydrological regime of Polish rivers is the seasonality of runoff associated with the occurrence of cold (winter) and warm (summer) seasons within a hydrological year, a seasonal approach is applied to describe specific seasonal features that can be masked when using annual data. In general, the results confirm popular impressions about changes in winter season runoff characteristics, i.e., significantly decreasing daily maxima, increasing daily minima and a decrease in concentration, and so a bigger uniformity of winter daily flows. An interesting behaviour of minimum flows in the summer has been revealed, which is contrary to social perceptions and the alarming changes taking place in the other parts of the world. Additionally, precipitation indexes related to the formation of droughts show no trends, e.g., the mean value of the maximum dry spell length.

**Keywords:** Vistula basin; runoff; precipitation; climate change

## 1. Introduction

The Vistula, not without reason, is called the queen of Polish rivers. It has been attracting the attention of researchers for centuries. So many words on the Vistula have been told and written in multiple books, presentations, scientific papers, reports and popular publications, [1–6] to cite only a small number of them, that it seems that everything about it is known and that it is difficult to say anything new. However, growing human pressures on the environment and especially on water resources along with global warming and its impacts on the water cycle require constant monitoring of changes in precipitation and the river's hydrological regime in order to assess the situation and point out harbingers of possible threats. This is usually done by increasing the number of observations and by extending the scope of the analysed characteristics and research tools used.

Water shortages are a growing concern in the world as water demand is growing. Therefore, it is crucial to understand how climate change and human impact affect water supplies. Svensson et al. [7] presented trend evaluation of 7-day and 30-day minimum flow at 21 stations worldwide, but no general pattern was found. In more recent studies, it was found that drought evolution is subject to the influence of not only the amount of precipitation but also the amount of soil moisture in the ground and variations in catchment water balance [8]. Droughts in 2003 and 2015 in Europe were studied by Van Lannen et al. [9]. The processes involved in a transition of meteorological drought into hydrological drought for the same two drought events in 2003 and 2015 were analysed by Laaha et al. [10]. In both cases, the influence of local conditions on drought development was underlined. Those recent drought events, followed by the 2018 drought, made European governments aware of the necessity of increasing resilience to droughts. Somorowska [11] presented changes in drought conditions in Poland during the past six decades using the

Standardized Precipitation Evaporation Index (SPEI). Hansel et al. [12] presented seasonal trends of very low precipitation in spring (April and June) and heavy precipitation trends in March and September. In general, both drought and heavy precipitation events are becoming more frequent in west-central Europe. However, there is no clear pattern seen in the development of drought events in Poland. Despite many papers describing the evolution of drought processes in Europe and in Poland in particular, most of them apply standardized indices. There is a lack of assessment of long-term trends of variables directly characterizing extreme events such as magnitudes of daily maxima and minima. The application of standardized indices is useful from the point of view of comparison between different catchments [13,14], but the physical meaning of a variable is lost and its trends are distorted. In addition, error is added due to the transformation performed and that error will be the largest for the least frequent variables such as extrema. In order to obtain a more clear picture of the processes involved, we decided to perform the analysis without the application of a standardization technique.

The research objectives was to answer the following questions. Firstly, is the combined impact of climate change and human activities recognized in the long-term series of annual and seasonal precipitation and flow data in the Vistula basin and what form does that impact take? Secondly, which aspects of the rainfall-flow regime are most affected in the Vistula basin and does the quantitative assessment support the general public perception? The third research question is to assess the joint trends in the mean and standard deviation of seasonal flow minima and the other selected characteristics in order to see whether those changes occur in parallel and how they are being transformed along the river.

The present study was carried out within the framework of the Chinese–Polish HUM-DROUGHT project: Human and Climate Impacts on Drought Dynamics and Vulnerability launched at the end of 2019. This paper is a continuation of a series of articles on droughts in the Vistula basin and especially along the Vistula course, namely [15,16], in which, among others, the analysis of the temporal and spatial variability of a number of drought indices was presented and discussed. Here, we have aimed at a statistical description of time changes of multiple characteristics of runoff in the entire period 1951–2018 and precipitation starting from 1952 to 2018 with two shorter data series up to 2014.

Since a concise description of the Vistula has been presented in [15], we will not repeat it here, although some information will be recalled for the sake of clarity of the text.

## 2. Study Area and Runoff and Precipitation Data

The study area is the River Vistula and its basin down to the last hydrological station located close to the mouth of the river—Tczew. The traditional hydrographic division of the basin encompasses the Upper Vistula basin to the San (from its upper part, the so-called Little Vistula basin, down to the River Przemsza), the San basin, the Middle Vistula down to the Narew mouth, the Narew basin, and downstream the Lower Vistula (see Figure 1). Due to the important human pressures on the hydrological regime from industry, especially the coal mining industry, the Little Vistula basin and the Przemsza basin in Figure 1 are linked together, showing the most affected part of the basin.

The sub-basins of the Vistula basin differ in orography, precipitation amount and structure, land cover and use, industry pressures on surface and ground waters, and many other aspects [17]. The hydrological regime also changes with the river course. From pluvial-nival in the upper part (down to the Jawiszowice station) through nival-pluvial downstream to Toruń and moderately-developed nival to the Vistula mouth [18]. The surface water resources in comparison to the area of the sub-basins of the Vistula presented in Figure 2 show the distribution of the mean annual runoff volume and indicate the basins with great water potential (the Upper Vistula and the San basin).

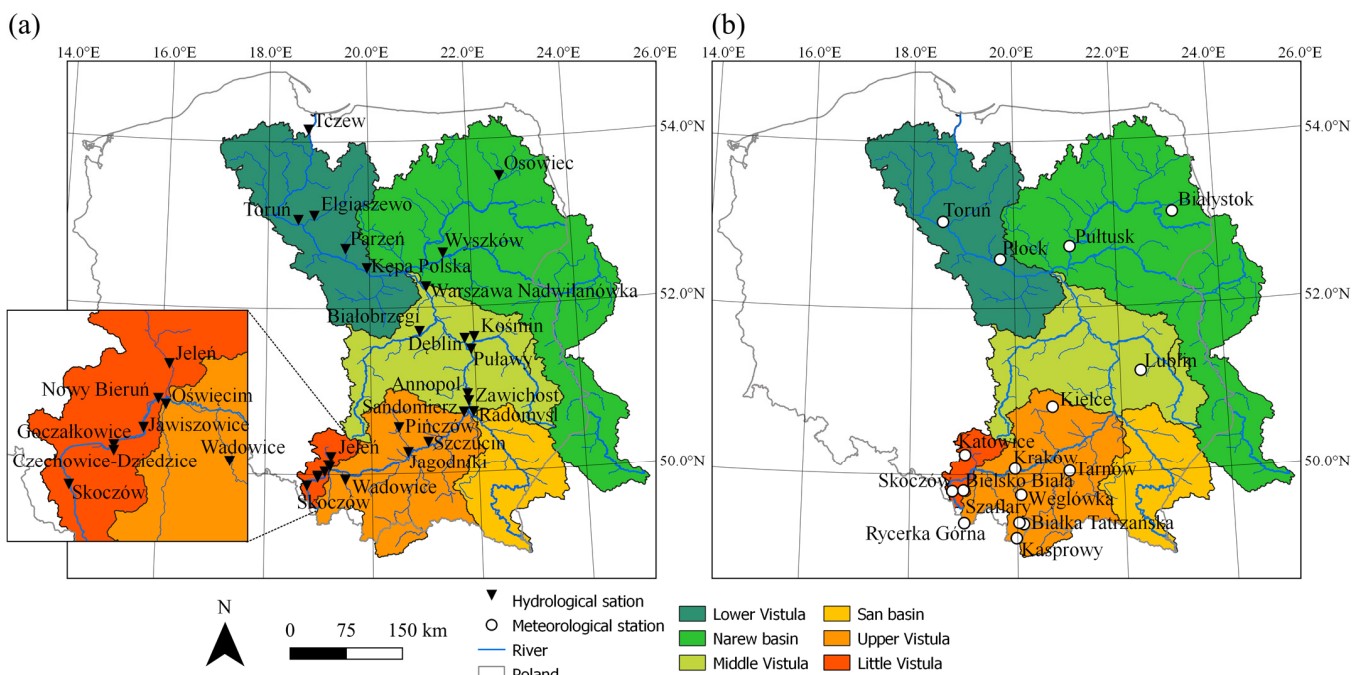

**Figure 1.** The hydrographic division of the Vistula basin with the hydrological (**a**) and precipitation (**b**) stations used in the study.

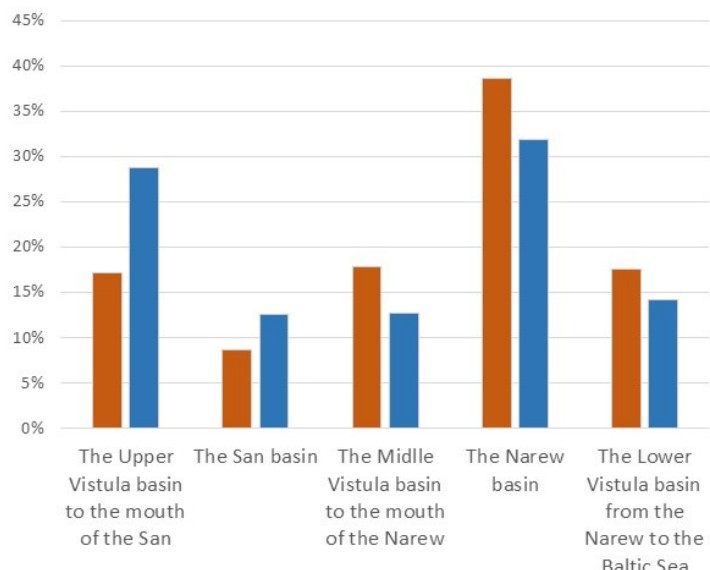

**Figure 2.** The runoff contribution of the Vistula sub-basins to the total runoff to the Baltic Sea. (Red) the share of the area in the total area of the Vistula basin; (blue) the share of the mean total outflow in the mean total outflow of the Vistula.

Poland has no conditions favourable for the construction of dams and water storage reservoirs. There are currently 61 reservoirs with an area larger than 10 ha and important barrages in the Vistula basin. Their total capacity is approx. 2860 million m$^3$, which is only 8.6% of the mean total annual runoff to the Baltic Sea. The increase of the total reservoir's capacity is depicted in Figure 3. Most of the reservoirs perform flood protection functions and ensure environmental flow downstream of the reservoir. They play a role in the water supply for the population and industry, hydro-energy production, and recreation.

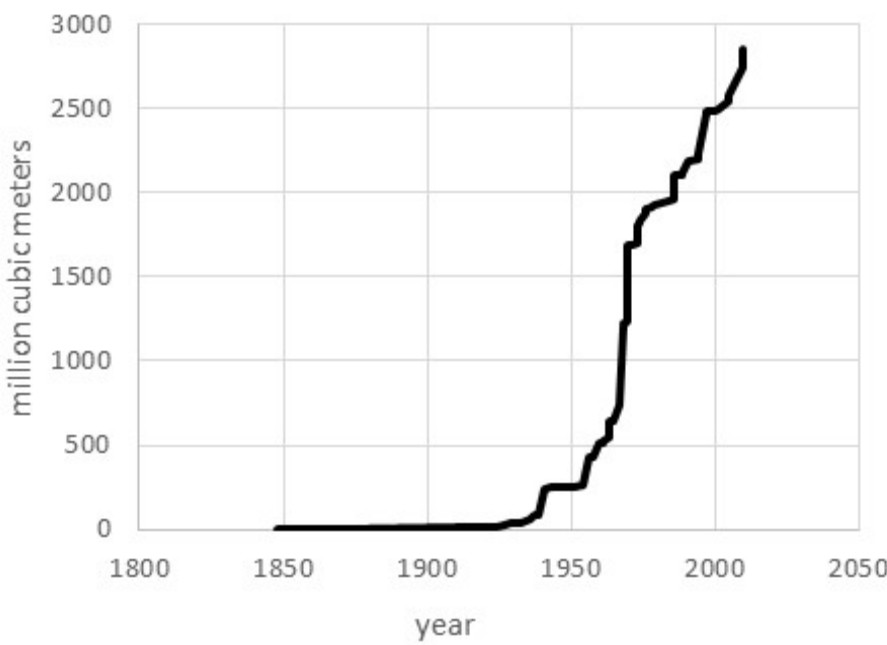

**Figure 3.** Cumulative total capacity of reservoirs in the Vistula basin.

There are three reservoirs on the Vistula's main stream. The first, Wisła-Czarne, put into operation in 1973, has a total capacity of 5.06 million m$^3$ and is located in the upstream part of the River Vistula, just after the two streams called the White and the Black Vistula form the watercourse. The second—Goczałkowice—launched in 1956 with a total capacity 165.5 million m$^3$, is located in the Little Vistula reach. It performs water supply functions, causes a significant reduction in the negative effects of floods in the river valley and regulation of the flow in periods of drought. The third reservoir, located in Włocławek on the Lower Vistula is a water barrage with a retention capacity of 453.6 million m$^3$ built in 1970 with a hydropower plant (160.2 MW) and flood protection functions. Other human interventions in the basin are discussed in [15].

It should be noted that the main feature of the hydrological regime of Polish rivers is a sequential occurrence of wet and dry periods lasting one to a dozen years [19,20]. The differences between mean annual flows during series of wet and dry years are large and statistically significant. Another important characteristic of the hydrological regime is the seasonality of runoff associated with the occurrence of cold and warm seasons within a hydrological year. Traditionally, the hydrological year (1 November–31 October) is divided into two half years: cool (winter) from 1 November to 30 April and warm (summer) from 1 May to 30 October. This division takes into account not only the temperature characteristics or the precipitation type (snow, snow contributed by rainfall), but also the vegetation cover and soil conditions that determine retention and evapotranspiration. The seasonal characteristics of flow differ, so their statistical properties ought to be analysed separately [15]. It is noteworthy that an interesting method of hydrological season delineation with respect to the river discharges and the base flow was presented recently in [21]. It was done by dividing the daily discharge transformation into three series, reflecting three statistical features estimated for particular days of the year from a multiyear average value, variation coefficient, and autocorrelation.

The choice of hydrological and precipitation gauging stations for further analysis was dictated by data availability in the full observation period, i.e., 1951–2018 (68 years). However, only a limited number of stations fulfil this requirement. The situation on the River Vistula is rather good, but the datasets at the stations on the outlet sections of the Vistula tributaries are generally about 20 years shorter. That is why the stations located on the tributaries do not cover all important inflows to the Vistula and some upstream

stations have been chosen instead as they can reveal the trends in the Vistula basin for smaller basin sizes.

The precipitation data in the Vistula basin are of poor quality from the point of view of the length of the observation period and completeness of the records. In addition, the data are stored by calendar time units. We have recalculated the precipitation amounts and other precipitation characteristics into the hydrological annual and seasonal time scales. This procedure results in the loss of one winter season, and in consequence one year of observation, i.e., the hydrological year 1951. The series of summer daily precipitation characteristics cover the full observation period. Two stations (Tarnów and Płock) have partially incomplete data. Observations of the precipitation type and snow cover data were terminated in 2014. A list of stations with basic information on precipitation in the observation period is given in Tables 1 and 2.

**Table 1.** List of the selected meteorological stations with essential information on mean seasonal and annual precipitation characteristics.

| No. | Station | Type [1] | Elevation (m a. s. l.) | Observation Period | Mean Precipitation Total (mm) | | | Mean Share of Winter in the Annual Total (%) | Mean Share of Snowfall in the Annual Total (%) |
|---|---|---|---|---|---|---|---|---|---|
| | | | | | Winter (Nov–Apr) | Summer (May–Oct) | Year (Nov–Oct) | | |
| 1 | Skoczów | SHM | 296 * | 1952–2018 | 330.8 | 600.5 | 931.3 | 35.5 | N/A |
| 2 | Bielsko-Biała | SHM | 398.89 | 1952–2018 | 325.7 | 663.5 | 989.2 | 32.9 | 20.8 |
| 3 | Katowice | P | 279.92 | 1952–2018 | 261.9 | 454.2 | 716.1 | 36.6 | 20.7 |
| 4 | Rycerka Górna | P | 668 * | 1952–2018 | 503.2 | 723.5 | 1226.7 | 41.0 | 31.0 |
| 5 | Węglówka | K | 490 * | 1952–2018 | 361.0 | 649.3 | 1010.3 | 35.7 | 24.9 |
| 6 | Kraków | WOM | 302 * | 1952–2018 | 231.6 | 454.3 | 685.9 | 33.8 | 17.6 |
| 7 | Kasprowy | WOM | 1988.75 | 1952–2018 | 718.3 | 1046.9 | 1764.9 | 40.7 | 61.9 |
| 8 | Szaflary | P | 652 * | 1952–2018 | 287.8 | 583.5 | 871.3 | 33.0 | 26.8 |
| 9 | Białka Tatrzańska | P | 733 * | 1952–2018 | 284.9 | 560.8 | 845.7 | 33.7 | 26.4 |
| 10 | Tarnów | K | 205 * | 1952–2014 | 234.3 | 474.5 | 708.8 | 33.1 | 18.7 |
| 11 | Kielce | SHM | 661.11 | 1952–2018 | 238.3 | 392.7 | 631.0 | 37.8 | 21.8 |
| 12 | Lublin | SHM | 339.7 | 1952–2018 | 216.0 | 373.6 | 589.6 | 36.6 | 21.9 |
| 13 | Białystok | SHM | 152.05 | 1952–2018 | 215.4 | 380.0 | 595.4 | 36.2 | 21.9 |
| 14 | Pułtusk | K | 88 * | 1952–2018 | 205.8 | 354.5 | 560.3 | 36.7 | 17.0 |
| 15 | Płock | SHM | 99 * | 1952–2014 | 195.4 | 336.6 | 532.0 | 36.7 | 17.4 |
| 16 | Toruń | SHM | 70.22 | 1952–2018 | 184.7 | 350.2 | 534.9 | 34.5 | 16.1 |

Note: [1] SHM—hydro-meteorological station (synoptic); WOM—Alpine meteorological observatory; K—climatological station; P—precipitation station. * Meta data not available in the public data repository, approximate estimation.

In general, the precipitation measurement scopes differ in different types of stations. The best observation scope, quality and completeness are provided by SHM stations and a WOM observatory with professional staff. So-called climatological and precipitation stations sometimes do not reach this level.

The list of hydrological stations used in the study is presented in Table 3. They are shown on a background of a simplified, schematic hydrographical river network structure. Precipitation gauging stations are assigned to the catchment area in which they are located. The major tributaries of the Vistula are marked even though they are not monitored by the hydrological station selected for analysis because of the length of observation series.

**Table 2.** Structure of daily precipitation; mean annual number of days in classes of precipitation amount in the observation period (see Table 1).

| No. | Station | Classes of Daily Precipitation (mm) | | | | | | | |
|-----|---------|------|------|-----------|-----------|-----------|------------|-------------|-------|
| | | 0.0 | 0.1 | (0.1;0.5> | (0.5;1.0> | (1.0;5.0> | (5.0;10.0> | (10.0;20.0> | >20.0 |
| 1 | Skoczów | N/A | 8.3 | 29.8 | 18.6 | 70.2 | 29.9 | 17.0 | 8.8 |
| 2 | Bielsko-Biała | 35.8 | 10.6 | 26.3 | 20.2 | 67.6 | 28.2 | 18.9 | 9.7 |
| 3 | Katowice | 43.1 | 14.4 | 29.8 | 22.0 | 67.2 | 24.8 | 13.8 | 4.9 |
| 4 | Rycerka Górna | 5.3 | 4.5 | 21.9 | 16.9 | 69.6 | 36.6 | 28.1 | 12.0 |
| 5 | Węglówka | 2.5 | 7.0 | 23.3 | 15.4 | 67.1 | 31.8 | 19.7 | 9.6 |
| 6 | Kraków | 42.5 | 13.4 | 31.8 | 20.7 | 65.1 | 23.0 | 13.1 | 4.8 |
| 7 | Kasprowy | 20.7 | 7.7 | 23.6 | 19.4 | 77.5 | 42.5 | 34.6 | 21.5 |
| 8 | Szaflary | 7.3 | 1.5 | 19.7 | 18.1 | 71.1 | 28.7 | 16.7 | 6.7 |
| 9 | Białka Tatrzańska | 14.6 | 10.4 | 22.6 | 15.9 | 73.0 | 29.0 | 17.3 | 6.3 |
| 10 | Tarnów | 35.3 | 13.3 | 29.7 | 21.9 | 64.1 | 23.4 | 12.9 | 5.7 |
| 11 | Kielce | 52.6 | 16.7 | 31.4 | 20.4 | 67.1 | 24.1 | 11.2 | 3.6 |
| 12 | Lublin | 49.7 | 16.6 | 32.5 | 21.2 | 64.7 | 21.9 | 9.7 | 3.6 |
| 13 | Białystok | 52.6 | 14.7 | 29.4 | 20.4 | 65.6 | 23.6 | 10.0 | 3,1 |
| 14 | Pułtusk | 15.1 | 6.9 | 23.7 | 19.3 | 59.1 | 20.5 | 9.9 | 3.3 |
| 15 | Płock | 52.0 | 15.2 | 30.5 | 20.7 | 63.6 | 20.2 | 9.2 | 2.6 |
| 16 | Toruń | 51.1 | 15.9 | 29.7 | 20.8 | 63.4 | 19.8 | 8.3 | 3.1 |

**Table 3.** Hydrological stations in the Vistula basin used in the research on the background of simplified schematic hydrographical network structures.

| No. | Hydrological Station on the Vistula | RIVER REACH/River | Hydrological Station on the Tributary | Total Area of the Tributary Basin | Tributary Side | Km | Area (km²) | Human Pressure on Natural Regime | Precipitation Stations |
|---|---|---|---|---|---|---|---|---|---|
| 1 | Skoczów | LITTLE VISTULA | | | | 71.1 | 296.7 | Quasi-natural | Skoczów |
| | | Iłownica | Czechowice-Dziedzice | 201.1 | Right | 1.5 | 193.9 | Altered | |
| 2 | Goczałkowice | LITTLE VISTULA | | | | 37.8 | 738.1 | Totally altered | |
| | | Biała | | 139.1 | Right | | | Totally altered | Bielsko-Biała |
| 3 | Jawiszowice | | | | | 23.7 | 970.6 | Altered | |
| 4 | Nowy Bieruń | LITTLE VISTULA | | | | 3.6 | 1747.7 | Altered | |
| | | Przemsza | Jeleń | 2121.5 | Left | 12.8 | 1995.9 | Totally altered | Katowice |
| | | Soła | Oświęcim | 1390.6 | Right | 3.0 | 1386.0 | Totally altered | Rycerka Górna |
| | | Skawa | Wadowice | 1160.1 | Right | 21.1 | 835.4 | Quasi-natural | |
| | | Raba | | 1537.1 | Right | | | Altered | Węglówka |
| 5 | Jagodniki | UPPER VISTULA | | | | 153.1 | 12,058.2 | Altered | Kraków |
| | | Dunajec | | 6804.0 | Right | | | Altered | Kasprowy Wierch. Szaflary. Białka. |
| | | Nida | Pińczów | 3862.0 | Left | 56.8 | 3352.5 | Natural | Tarnów Kielce |
| 6 | Szczucin | UPPER VISTULA | | | | 194.1 | 23,900.6 | Altered | |
| | | Wisłoka | | 4110.2 | Right | | | Quasi-natural | |
| 7 | Sandomierz | UPPER VISTULA | | | | 268.4 | 31,846.5 | Altered | |
| | | San | Radomyśl | 16,861.3 | Right | 10.3 | 16,823.8 | Altered | |
| 8 | Zawichost | MIDDLE VISTULA | | | | 287.6 | 50,731.8 | Altered | |
| 9 | Annopol | MIDDLE VISTULA | | | | 298.4 | 51,518.1 | Natural | |
| 10 | Puławy [1] | MIDDLE VISTULA | | | | 372.5 | 57,263.6 | Natural | |
| | | Wieprz | Kośmin | 10,415.2 | Right | 17.9 | 10,230.6 | Quasi-natural | Lublin |
| 11 | Dęblin | MIDDLE VISTULA | | | | 393.7 | 68,234.3 | Natural | |
| | | Pilica | Białobrzegi | 9273.0 | Left | 45.3 | 8664.2 | Quasi-natural | |
| 12 | Warszawa Nadwilanówka [2] | MIDDLE VISTULA | | | | 503.5 | 84,539.5 | Natural | |

**Table 3.** *Cont.*

| No. | Hydrological Station on the Vistula | RIVER REACH/River | Hydrological Station on the Tributary | Total Area of the Tributary Basin | Tributary Side | Km | Area (km²) | Human Pressure on Natural Regime | Precipitation Stations |
|---|---|---|---|---|---|---|---|---|---|
| | | Narew | | 75,175.2 | Right | | | Totaly altered | Białystok Pułtusk |
| | | | Osowiec/ Biebrza | 7057.4 | Right | 50.3 | 4365.1 | Natural | |
| | | | Wyszków/ Bug | 39,420.3 | Left | 33.8 | 39,119.4 | Natural | |
| | | Bzura | | 7787.5 | Left | | | Quasi-natural | |
| 13 | Kępa Polska [3] | LOWER VISTULA | | | | 606.5 | 168,956.1 | Natural | Płock |
| | | Skrwa (right) | Parzeń | 1704.0 | Right | 20.8 | 1534.2 | Quasi-natural | |
| 14 | Toruń | LOWER VISTULA | | | | 734.7 | 181,033.4 | Altered | Toruń |
| | | Drwęca | Elgiszewo | 5343.5 | Right | 25.8 | 4959.4 | Natural | |
| 15 | Tczew | LOWER VISTULA | | | | 908.6 | 194,376.0 | Altered | |

Note: [1] Change in location in 2004. [2] Change in location in 1968. [3] Change in location in 1969.

## 3. Methodology

### 3.1. Characteristics of Daily Flow and Precipitation Processes

Since the objective of this research is to assess temporal changes in different aspects of runoff and precipitation processes, a wide set of statistical measures and indicators was defined and applied. They were chosen to quantify the main features of seasonal variability of flow and precipitation. The list of measures and indicators applied is given in Table 4.

**Table 4.** The list of statistical measures and indicators used in this study.

| Aspect | Index | Description | Unit |
|---|---|---|---|
| **High flows** | Max | Magnitude of seasonal daily maximum flow | $[m^3/s]$ |
| | Duration | Number of days with the flow over a threshold | [days] |
| **Low flows** | Min | Magnitude of seasonal daily minimum flow | $[m^3/s]$ |
| | Duration | Number of days with the flow below a threshold. | [days] |
| **Timing** | T of max | Number of the day when the highest flow occurred | - |
| | T of min | Number of the day when the lowest flow occurred | - |
| | Centr. (Centroid) | Centroid of seasonal hydrograph with respect to time | [days] |
| | Median | Number of the day when the half of seasonal runoff is achieved | [days] |
| **Runoff** | Volume | Volume of the seasonal runoff | $[m^3]$ |
| **Concentration of daily flows** | Inertia | Moment of inertia of dimensionless seasonal hydrograph with respect to the time coordinate of the centroid | $[day^2]$ |
| | Gini | Gini index calculated for seasonal daily flows | - |
| **Precipitation amount** | Seasonal and annual total precipitation amount | Sum of precipitation in seasonal and annual time scales | [mm] |
| | Annual totals of rainfall and snowfall | Annual rainfall and snowfall sum of precipitation | [mm] |
| | Precipitation totals in September and October | Monthly sums of precipitation in September and October | [mm] |
| | Share of snowfall in the annual total | Fraction of snowfall in annual total | [%] |
| **Number of days with precipitation** | Number of days with precipitation | Seasonal and annual number of days with precipitation | [days] |
| | Annual number of days with rain and with snow | Number of days with precipitation in seasonal and annual time scales | [days] |
| **Concentration of daily precipitation** | Gini | Gini index calculated for seasonal daily precipitation (zero values included) | - |
| **Snow cover** | Maximum thickness of snow cover | Annual maximum thickness of snow cover | [cm] |
| | Number of days with snow cover | Annual number of days with snow cover | [days] |
| **Dry periods** | Maximum dry spell length | Maximum number of consecutive days with precipitation not greater than 0.1 mm, i.e., without precipitation, precipitation trace or precipitation equal to 0.1 mm. Due to the importance of summer–autumn low flow, the maximum is determined in the annual periods starting from April 1st to March 31st | [days] |
| **Daily precipitation structure** | - | Number of days with precipitation in class intervals in a seasonal time frame | - |

For the runoff process, the indexes cover five aspects: high flows, low flows, timing, the runoff volume and the concentration of daily flows. High and low flows are characterized by the magnitude of daily maxima and minima, respectively, and the number of days with flow over/under the specified threshold (the mean of seasonal maxima and minima, respectively). However, it should be stressed that the use of the fixed dates of the beginning and the end of seasons and hydrological years is not a fully process-oriented approach and introduces some formalism to the analysis. The rigid time frames, very useful and needed from the practical point of view, and widely used all over the world, can crosscut the high or low flow periods in some years. In our detailed investigations of hydrological droughts, a "drought year" is defined on the basis of the dates of the minima. In an overwhelming number of cases, using 1st April as the beginning of the "drought year" is an appropriate solution. This problem will be tackled in the next publication on hydrological droughts, where the process-oriented approach will be applied.

The timing aspect is described by four characteristics: time of maxima and minima, hydrograph centroid, location of the median point on the time axis corresponding to the time in which the half of the runoff volume is achieved. The runoff volume does not need any explanation. The concentration of daily flows is measured by two characteristics: the moment of inertia of the hydrograph and the Gini index.

The precipitation process is characterized by the total precipitation amount, the number of days with precipitation, rainfall total, snowfall total, the number of rainy days and days with snow, the share of snowfall amount in the annual precipitation amount, the number of days with snow cover and the annual maximum thickness of snow cover, the concentration of daily precipitation (Gini index), and the maximum dry spell and the precipitation totals in September and October (which is important for the development of the summer–autumn low flows). The structure of daily precipitation amounts in a range of classes was also analysed.

Among the characteristics listed above, some require further explanation. These are the centroid and the inertia of daily flow hydrograph and the Gini index of concentration [22], which are rarely used in hydrological research. A modification of the Gini method was proposed and applied in [23] where daily precipitation concentrations across Europe in the period of 1971–2010 were investigated.

The centroid and the moment of inertia and its square root—the radius of gyration—were introduced to describe the shape of the hydrograph with respect to time. The hydrograph is treated here as a rigid body and the time coordinate of the centroid is found. Then, the moment of inertia with respect to the vertical axis passing through the centroid is calculated. The above definitions have the same interpretation as the mean value, the variance, and the standard deviation calculated for a random variable. Their changes are expected to measure the time shifts of the main body of the hydrograph and the de-concentration of the centroid. It is expected that the two characteristics will be able to quantify the changes in the hydrograph shape, such as the change of one important flood to a number of rather small flood waves. To ensure the comparability of the inertia scores in different years, the dimensionless hydrograph has been used, i.e., all daily flows were divided by their total.

The Gini index (or coefficient) measures the extent to which the distribution of the variable deviates from a uniform distribution, so it gives the concentration measure of the distribution. A big advantage of this measure is that the Gini index values range from zero (uniform distribution) to 1 (all values are concentrated in one point). The index is directly related to the so-called Lorenz curve [24]. The curve shows the proportion of the total runoff volume in a season assumed by the bottom d% of days. Without the loss of generality, one can use daily flows and their sum in place of runoff volumes to simplify the calculations. An exemplary plot of the Lorenz curves from Zawichost station is depicted in Figure 4.

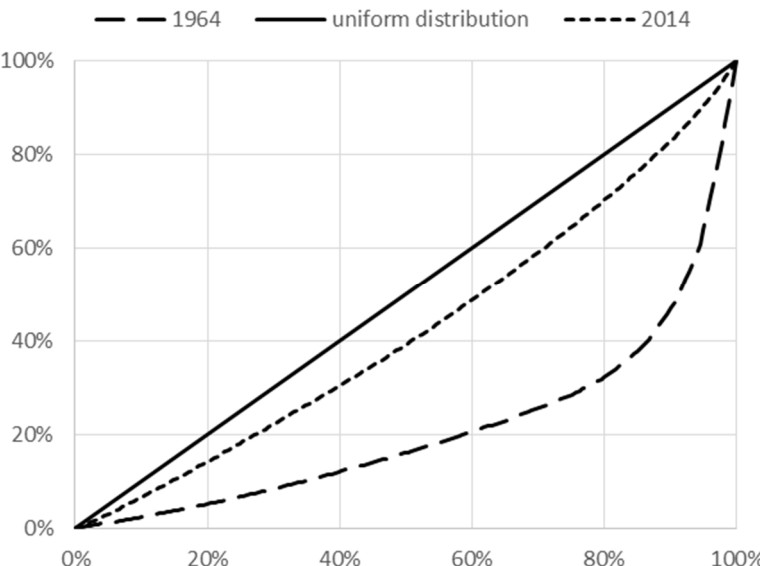

**Figure 4.** Lorenz curves for Zawichost hydrological station for the winter of 1964 and 2014. Ordinate (y) axis—cumulative share of days from lowest to largest daily flow. Abscissa (x) axis—cumulative share of daily flow in the sum of flow.

The Gini index corresponds to twice the area between the line of uniform distribution and the Lorenz curve. In the case presented above, the Gini index was equal 0.577 in 1964 and 0.157 in 2014. In Figure 5, the corresponding hydrographs of daily flow are shown. The difference between hydrographs in 1964 and 2014 is spectacular, and so is the difference in concentration measured by the Gini index. The Gini index is commonly used in economics to describe the distribution of income or inequality of wealth distribution in different countries around the world, e.g., [25].

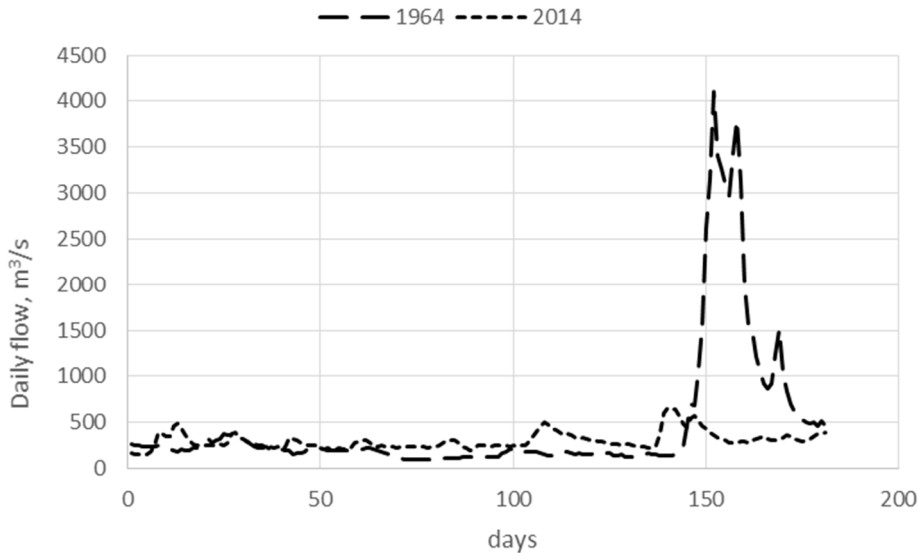

**Figure 5.** Hydrographs of daily flow in winter of 1964 and 2014.

*3.2. Methods of Analyses*

Climate change and its hydrological consequences can be considered significant or insignificant from the point of view of physical, environmental, economic, or statistical criteria. However, because there are no generally accepted physical, environmental, or economic approaches that would allow for the classification of such changes nor sufficiently detailed datasets, it was assumed that the method of research would be based only on statistical criteria.

The Mann sequential test [26] was used to detect trends in the characteristics of runoff and precipitation described in the previous section. The Mann test is a version of the Mann-Kendal test. It enables us to follow the values of the test statistics (forward), $uF$, over time and use the backward statistics, $uB$, to find the approximate time of the start of the trend. The basic assumption for the application of the test is that the random variables are independent and identically distributed (*iid*). The presence of serial correlation (autocorrelation) in hydro-meteorological time series affects the variance of the test statistics and often makes the detection of deterministic gradual or abrupt changes problematic [27]. In hydrological practice, annual or seasonal data are often assumed to be *iid* random variables without checking the autocorrelations (last year's snow does not count). However, the main feature of the hydrological regime of Polish rivers—a sequential occurrence of wet and dry periods described in [15]—makes us question this a priori assumption. The autocorrelation in the series of characteristics used in this study revealed that 16% of the proposed runoff characteristics along the Vistula's course suffer from significant autocorrelation. The share of significantly autocorrelated series on the tributaries is even greater and reaches about 24%. The main characteristics of precipitation show the significant autocorrelation only in about 9% of characteristics.

There is a rich literature on the issue of the impact of autocorrelation on the results of trend tests, e.g., [27–31], showing that the violation of the assumption of independence can significantly affect the value of the type I error. Type I errors are generally inflated by positive and deflated by negative autocorrelation. However, it is noteworthy that Yue & Wang [27] found that the existence of a trend in a time series produces a spurious serial correlation when there is no serial correlation, and the presence of a trend will increase the estimate of positive serial correlation when a serial correlation exists. To reduce this effect, pre-whitening procedures are recommended [27,28].

According to [28], the pre-whitening procedure is not needed for large samples ($n \geq 50$) and high slopes of a dimensionless trend ($b \geq 0.01$), where it would cause significant power loss if applied because serial correlation has a negligible effect on the rejection rate of the test in these cases. It should be applied, however, in other cases to prevent the detection of a non-existent trend. Pre-whitening will not cause a significant loss of power in these cases, with the possible exception of cases with very low values of the coefficient of variation, which can be estimated from a sample with a small sampling error when its population value is small.

We applied this recommendation in the detection of changes in the test statistics values. No pre-whitening procedure has been applied. In Tables 5–8 (seasonal flow characteristics), we have marked the direction of change of the test statistics $uF$ when the autocorrelation is taken into account. The marks represent the diagonal lines from the upper left corner to the lower right to show that the value can decrease if autocorrelation is taken into account and the opposite for the cases when the autocorrelation causes an increase in the presented value. No marks have been applied in Tables 9–13 with the results for precipitation because the possible increases/decreases in the values of the test statistics do not influence the general view of the results.

The third research problem marked in the introduction touches the assessment of trend in the mean value and standard deviation of instantaneous seasonal flow minima and how they are transformed along the course of the Vistula. The method used for the assessment is described in [32–34]. It involves the simultaneous estimation of trends in mean value and standard deviation by means of the weighted least squares (WLS) method. Trend studies are subject to numerous uncertainties and assumptions. Due to numerous sources of uncertainty in trend detection, the form of trend used in non-stationary analysis ought to be the simplest possible due to the parsimony principle. A linear trend in the mean and the standard deviation is generally recommended. It is also important to use, if necessary and possible, the seasonal data in trend analysis rather than annual ones, which can sometimes fail to reveal the full temporal complexity of flow and precipitation

trends [35]. Moreover, the seasonal values are more susceptible to the climate variability and change and the annual approach can mask important changes in seasonal dynamics.

**Table 5.** Results of the Mann test (*uF* statistics) for trends in winter season daily flow characteristics along the Vistula course; orange colour denotes positive *uF* values, blue colour negative. The marks represent the diagonal lines from the upper left corner to the lower right to show that the value can decrease if autocorrelation is taken into account and the opposite is true for the cases when the autocorrelation causes an increase in the presented value. Numbers in bold indicate statistically significant changes at α level 0.05.

| No. | Station | High Flows | | Low Flows | | Timing | | | | Runoff | Concentration | |
| --- | --- | --- | --- | --- | --- | --- | --- | --- | --- | --- | --- | --- |
| | | Max | Duration | Min | Duration | Time of Max | Time of Min | Centr. | Median | Volume | Inertia | Gini |
| 1 | Skoczów | 0.28 | 0.77 | **2.38** | −1.84 | −1.08 | −1.58 | −1.43 | −1.64 | 1.57 | −0.12 | −0.52 |
| 2 | Goczałkowice | −1.23 | −1.91 | 0.21 | −0.96 | −1.32 | 0.12 | −1.10 | −0.95 | 0.32 | 0.29 | −1.88 |
| 3 | Jawiszowice | −1.92 | **−2.23** | 1.15 | −0.83 | −0.56 | 0.36 | −1.08 | −1.17 | 0.32 | 0.81 | **−2.19** |
| 4 | Nowy Bieruń | −1.37 | −1.46 | **3.65** | **−2.99** | −1.05 | 0.01 | −1.32 | −1.31 | 0.68 | 0.99 | **−2.37** |
| 5 | Jagodniki | **−2.12** | −1.88 | 1.36 | −1.08 | −0.88 | −0.61 | −1.37 | −1.24 | −1.02 | 0.33 | **−2.34** |
| 6 | Szczucin | −1.12 | −1.68 | **2.30** | −1.68 | −0.91 | 0.16 | −1.21 | −0.92 | 0.45 | 0.38 | −1.81 |
| 7 | Sandomierz | −1.99 | **−2.21** | **3.20** | **−2.07** | −1.16 | 0.36 | −1.36 | −1.19 | 0.38 | 0.58 | −1.96 |
| 8 | Zawichost | **−2.41** | **−2.54** | **2.10** | **−2.15** | −0.94 | −0.65 | −1.25 | −1.11 | −0.69 | 0.89 | **−2.13** |
| 9 | Annopol | **−2.08** | **−2.01** | 1.44 | −1.31 | −0.94 | −0.49 | −1.12 | −1.06 | −0.48 | 0.85 | −1.94 |
| 10 | Puławy | −1.97 | −1.86 | 1.84 | −1.27 | −0.55 | 0.29 | −1.04 | −0.88 | −0.47 | 0.71 | −1.79 |
| 11 | Dęblin | −1.88 | −1.68 | 1.55 | −1.51 | −0.88 | −0.07 | −1.21 | −1.14 | −0.43 | 0.50 | **−2.06** |
| 12 | Warszawa | −1.81 | −1.69 | **2.28** | **−3.55** | −1.28 | 0.14 | −1.32 | −1.29 | 0.06 | 0.39 | **−2.57** |
| 13 | Kępa Polska | **−2.47** | **−2.12** | **2.08** | **−2.46** | −0.84 | −0.77 | −1.36 | −1.77 | 0.35 | −0.95 | **−3.13** |
| 14 | Toruń | **−2.60** | **−2.23** | 1.33 | −1.55 | −0.88 | −1.71 | −1.38 | −1.54 | −0.17 | −0.84 | **−3.13** |
| 15 | Tczew | −1.94 | −1.45 | 1.60 | −1.75 | −0.67 | −1.69 | −1.45 | −1.53 | 0.15 | −0.94 | **−2.90** |

**Table 6.** Results of the Mann test (*uF* statistics) for trends in the summer season daily flow characteristics along the Vistula course; orange colour denotes positive *uF* values, blue colour negative. The marks represent the diagonal lines from the upper left corner to the lower right to show that the value can decrease if autocorrelation is taken into account and opposite is true for the cases when the autocorrelation causes an increase in the presented value. Numbers in bold indicate statistically significant changes at α level 0.05.

| No. | Station | High Flows | | Low Flows | | Timing | | | | Runoff | Concentration | |
| --- | --- | --- | --- | --- | --- | --- | --- | --- | --- | --- | --- | --- |
| | | Max | Duration | Min | Duration | Time of Max | Time of Min | Centr. | Median | Volume | Inertia | Gini |
| 1 | Skoczów | 0.08 | 0.65 | 1.29 | −0.86 | 0.19 | −0.25 | 1.35 | 0.05 | −1.00 | 1.38 | **2.01** |
| 2 | Goczałkowice | −1.78 | −1.84 | 0.65 | −0.96 | 0.41 | −0.49 | 1.57 | 1.18 | −0.94 | 1.41 | −0.17 |
| 3 | Jawiszowice | −1.53 | −0.33 | 0.2 | 0.65 | 0.29 | **−2.04** | 1.45 | 0.88 | −0.86 | 1.63 | −1.15 |
| 4 | Nowy Bieruń | −1.23 | −0.61 | **3.01** | **−2.69** | −0.16 | −1.27 | 1.61 | 1.2 | −0.37 | 1.38 | **−2.08** |
| 5 | Jagodniki | −0.09 | −0.68 | −0.70 | 1.42 | −0.84 | −0.68 | 0.56 | 0.45 | −1.30 | 1.76 | −0.36 |
| 6 | Szczucin | 0.39 | −0.76 | 1.56 | −0.75 | −0.56 | 0.48 | 0.59 | 0.77 | 0.21 | 1.05 | 0.15 |
| 7 | Sandomierz | 0.21 | −0.68 | 1.08 | −0.84 | 0.16 | −0.64 | 1.01 | 1 | 0.16 | 1.25 | 0.53 |
| 8 | Zawichost | −0.03 | −0.64 | 0.87 | −0.69 | 0.19 | −0.07 | 0.82 | 0.91 | −0.29 | 1.02 | 0.07 |
| 9 | Annopol | 0.12 | −0.12 | 0.24 | −0.04 | 0.63 | −0.57 | 0.91 | 0.67 | −0.10 | 1.01 | 0.28 |
| 10 | Puławy | 0.4 | −0.03 | 0.44 | −0.09 | 0.7 | −1.07 | 0.81 | 0.86 | −0.35 | 0.93 | 0.5 |
| 11 | Dęblin | 0.55 | 0.16 | 0.58 | −0.23 | 1.03 | −1.00 | 1.38 | 1.48 | −0.12 | 1.15 | −0.07 |
| 12 | Warszawa | 0.34 | −0.54 | 0.52 | 0.15 | 0.46 | −0.91 | 0.78 | 0.78 | −0.18 | 1.24 | 0.57 |
| 13 | Kępa Polska | 0.31 | −0.44 | 0.15 | 0.29 | 1.47 | −1.02 | 0.37 | 0.56 | −0.22 | 1.04 | 0.4 |
| 14 | Toruń | −0.36 | −0.38 | −1.29 | 1.14 | 0.63 | −0.20 | 0.66 | 0.57 | −0.47 | 1.02 | 0.1 |
| 15 | Tczew | −0.40 | −0.47 | 0.09 | 0.19 | 1.11 | 0.23 | 0.66 | 0.69 | −0.13 | 1.1 | −0.29 |

**Table 7.** Results of the Mann test (*uF* statistics) for trends in the winter season daily flow characteristics at hydrological stations on the Vistula tributaries; orange colour denotes positive *uF* values, blue colour negative. The marks represent the diagonal lines from the upper left corner to the lower right to show that the value can decrease if autocorrelation is taken into account and the opposite is true for the cases when the autocorrelation causes an increase in the presented value. Numbers in bold indicate statistically significant changes at α level 0.05.

| River/Station | High Flows | | Low Flows | | Timing | | | | Runoff | Concentration | |
|---|---|---|---|---|---|---|---|---|---|---|---|
| | Max | Duration | Min | Duration | Time of Max | Time of Min | Centr. | Median | Volume | Inertia | Gini |
| Iłownica/Cz.-Dz. | −1.21 | −1.34 | −0.07 | −0.23 | −1.84 | 1.16 | −1.46 | −1.45 | −0.75 | 0.58 | 0.12 |
| Przemsza/Jeleń | **−3.63** | **−2.77** | 0.88 | −0.56 | −0.90 | −0.49 | −0.65 | −1.09 | −0.73 | −0.63 | **−3.22** |
| Soła/Oświęcim | −1.28 | −1.63 | 0.57 | 0.33 | −1.42 | −0.14 | −1.70 | −1.50 | 0.28 | −0.12 | −0.72 |
| Skawa/Wadowice | 0.40 | 0.55 | 0.07 | −0.23 | −1.31 | −0.59 | −0.43 | 0.21 | 1.14 | 0.22 | 0.83 |
| Nida/Pińczów | **−3.37** | **−3.02** | 1.37 | −0.94 | 0.16 | −1.30 | −0.39 | −0.42 | −1.24 | 1.26 | **−2.80** |
| San/Radomyśl | **−1.99** | **−2.09** | 4.77 | **−4.18** | 1.14 | −1.31 | −0.78 | −0.90 | 0.90 | 0.74 | **−3.62** |
| Wieprz/Kośmin | −1.73 | −1.04 | 3.64 | **−2.73** | −0.27 | 0.02 | −1.29 | −1.79 | 1.67 | −0.19 | **−3.04** |
| Pilica/Białobrzegi | **−4.00** | **−3.10** | 2.15 | −1.74 | −1.24 | −0.48 | −1.12 | −1.22 | −0.60 | 1.09 | **−3.68** |
| Biebrza/Osowiec | −1.24 | −1.13 | 3.85 | **−3.54** | **−2.35** | −1.41 | −1.43 | **−2.42** | 2.36 | **−3.71** | **−3.11** |
| Bug/Wyszków | **−2.49** | **−2.43** | 3.33 | **−3.00** | −0.49 | −1.31 | **−2.03** | **−2.49** | 0.42 | −0.36 | **−3.71** |
| Skrwa/Parzeń | −0.58 | −0.50 | 0.28 | −0.16 | −1.00 | **−2.15** | −0.22 | −0.99 | 0.18 | −1.36 | −0.54 |
| Drwęca/Elgiszewo | 0.16 | 0.10 | 0.26 | −0.96 | −0.57 | −1.52 | −0.21 | −0.02 | 0.43 | **−1.98** | −0.63 |

**Table 8.** Results of the Mann test (*uF* statistics) for trends in the summer season daily flow characteristics at hydrological stations on the Vistula tributaries; orange colour denotes positive *uF* values, blue colour negative. The marks represent the diagonal lines from the upper left corner to the lower right to show that the value can decrease if autocorrelation is taken into account and the opposite is true for the cases when the autocorrelation causes an increase in the presented value. Numbers in bold indicate statistically significant changes at α level 0.05.

| River/Station | High Flows | | Low Flows | | Timing | | | | Runoff | Concentration | |
|---|---|---|---|---|---|---|---|---|---|---|---|
| | Max | Duration | Min | Duration | Time of Max | Time of Min | Centr. | Median | Volume | Inertia | Gini |
| Iłownica/Cz.-Dz. | −1.26 | −1.23 | −0.47 | 0.84 | 0.02 | −0.09 | **2.18** | 1.76 | 0.90 | 1.18 | −0.03 |
| Przemsza/Jeleń | −1.28 | −0.42 | −0.29 | 0.71 | 0.52 | 0.22 | 0.53 | 0.21 | 0.90 | 1.36 | −1.88 |
| Soła/Oświęcim | −0.61 | −0.19 | −0.63 | 1.22 | −0.76 | **−2.19** | 1.01 | 0.33 | −1.35 | 1.46 | 0.70 |
| Skawa/Wadowice | 0.67 | 0.97 | **−2.78** | **2.23** | 0.97 | 0.20 | 0.41 | 0.73 | 0.05 | 1.35 | **2.42** |
| Nida/Pińczów | −1.29 | −0.45 | −1.50 | 1.87 | −1.11 | 0.87 | −1.62 | −1.26 | −1.29 | 1.81 | 0.04 |
| San/Radomyśl | −0.30 | −0.02 | **2.52** | **−2.09** | 0.18 | 0.51 | −0.01 | −0.46 | 0.70 | 0.45 | −1.18 |
| Wieprz/Kośmin | 0.32 | 0.76 | **2.02** | **−2.22** | 0.80 | 0.74 | 1.45 | 1.48 | **2.17** | 0.94 | −0.81 |
| Pilica/Białobrzegi | −1.61 | −0.52 | −1.07 | 0.94 | −0.66 | 1.51 | −0.66 | −0.55 | −1.28 | 1.82 | −0.65 |
| Biebrza/Osowiec | −1.45 | −1.23 | −0.48 | 0.15 | −1.35 | −0.47 | 1.03 | 1.00 | −0.75 | 1.52 | −1.07 |
| Bug/Wyszków | −0.21 | −0.02 | 1.51 | −1.21 | −0.38 | 0.49 | 0.35 | 0.42 | 0.86 | 0.60 | −1.00 |
| Skrwa/Parzeń | −0.41 | 0.64 | −1.69 | **2.37** | 0.28 | 0.84 | −1.43 | −1.86 | −1.47 | 1.04 | 1.57 |
| Drwęca/Elgiszewo | −0.84 | −0.46 | −1.52 | 0.86 | −1.04 | 1.67 | 0.16 | 0.00 | −1.24 | 1.24 | 0.29 |

　　An important problem in hydrological studies, and especially in the realm of trend detection, is the number of measurements and observations and their uncertainty. Most of the methods of trend analysis require complete data over the analysed period, but in practice such a situation is rather rare. Nowadays, new methods are applied for trend detection, e.g., least-squares spectral analysis (LSSA) to analyse unequally spaced time series in the frequency domain. [36]. The method also enables the detection of jumps and breakpoints and can show how the components of the data change over time (see also [21]) and how climate change may impact the streamflow over time. It is expected that spectral and wavelet analyses can also help to provide more reliable climate and hydrological forecasts. It should be mentioned here that in the case of complete data the trend assessments are almost the same as those obtained by traditional methods.

**Table 9.** Results of the Mann test (*uF* statistics) for trends in precipitation characteristics in the seasonal and annual time scales; orange colour denotes positive *uF* values, blue colour negative. Numbers in bold indicate statistically significant changes at α level 0.05.

| No. | Station | Total Precipitation Amount | | | Number of Days with Precipitation | | | Annual Total Amount | |
|---|---|---|---|---|---|---|---|---|---|
| | | Winter | Summer | Year | Winter | Summer | Year | Rainfall | Snowfall |
| 1 | Skoczów | 0.27 | 0.07 | −0.16 | 0.91 | **2.03** | **2.28** | N/A | N/A |
| 2 | Bielsko-Biała | −1.10 | 0.16 | −0.87 | 0.75 | 0.01 | 0.46 | 0.35 | **−2.86** |
| 3 | Katowice | 0.92 | −0.16 | 0.18 | 1.79 | 1.90 | **2.37** | −0.06 | −0.44 |
| 4 | Rycerka Górna | 0.24 | −0.55 | −0.58 | **3.53** | **3.71** | **3.70** | −0.38 | −0.54 |
| 5 | Węglówka | 1.07 | 1.08 | 1.27 | **3.98** | **3.89** | **4.41** | 0.71 | 0.47 |
| 6 | Kraków | 0.04 | 0.55 | 0.08 | 1.59 | −0.53 | 0.50 | 0.92 | −1.96 |
| 7 | Kasprowy | −1.78 | 1.62 | 0.29 | **−2.13** | −1.38 | **−3.07** | 1.46 | −1.06 |
| 8 | Szaflary | 0.14 | 1.64 | 0.56 | −0.40 | 1.60 | 0.29 | 1.43 | −1.59 |
| 9 | Białka Tatrzańska | −0.86 | 0.48 | 0.04 | 0.35 | **2.14** | 1.26 | 1.13 | **−2.19** |
| 10 | Tarnów | 0.01 | 1.52 | 0.88 | **3.46** | 1.77 | −0.75 | 1.78 | 1.76 |
| 11 | Kielce | −1.25 | 0.81 | −0.35 | **2.14** | **2.50** | **2.42** | 0.55 | **−3.32** |
| 12 | Lublin | 0.41 | 0.93 | 0.66 | 1.31 | 0.13 | 0.49 | 0.81 | −0.68 |
| 13 | Białystok | 0.07 | **1.97** | 0.82 | 1.24 | 0.80 | 1.33 | 1.91 | **−2.65** |
| 14 | Pułtusk | **2.02** | **2.26** | **2.28** | **−2.42** | **−1.31** | **−3.16** | 2.77 | −1.63 |
| 15 | Płock | 0.26 | −0.39 | −0.36 | **3.39** | **2.17** | **3.38** | 0.39 | **−2.85** |
| 16 | Toruń | 1.88 | 0.90 | 1.01 | 0.85 | 0.52 | 0.29 | 1.34 | −0.56 |

**Table 10.** Results of the Mann test (*uF* statistics) for trends in annual precipitation characteristics; orange colour denotes positive *uF* values, blue colour negative. Numbers in bold indicate statistically significant changes at α level 0.05.

| No. | Station | Annual Number of Days | | Share of Snowfall in the Annual Total | Maximum Thickness of Snow Cover | Number of Days with Snow Cover |
|---|---|---|---|---|---|---|
| | | with Rain | with Snow | | | |
| 1 | Skoczów | N/A | N/A | N/A | N/A | N/A |
| 2 | Bielsko-Biała | 0.77 | −0.55 | **−2.26** | −1.31 | **−3.09** |
| 3 | Katowice | **2.00** | 0.39 | −0.07 | −0.94 | **−2.22** |
| 4 | Rycerka Górna | **5.04** | −0.08 | −0.14 | N/A | N/A |
| 5 | Węglówka | **3.98** | 1.30 | 0.31 | N/A | N/A |
| 6 | Kraków | **2.38** | **−2.07** | −1.84 | −1.15 | **−1.99** |
| 7 | Kasprowy | −0.94 | −1.69 | −1.65 | −1.19 | **−2.17** |
| 8 | Szaflary | **2.32** | **−2.73** | −1.68 | N/A | N/A |
| 9 | Białka Tatrzańska | **3.07** | **−2.22** | **−2.20** | N/A | N/A |
| 10 | Tarnów | −1.61 | −0.52 | 0.50 | 0.07 | −0.58 |
| 11 | Kielce | 0.94 | 1.83 | **−2.77** | −1.28 | **−3.40** |
| 12 | Lublin | −1.72 | 1.90 | −0.89 | −0.01 | −1.30 |
| 13 | Białystok | −0.33 | 1.63 | **−3.30** | −0.99 | **−2.89** |
| 14 | Pułtusk | 0.98 | **−5.05** | **−2.74** | −0.54 | **−3.00** |
| 15 | Płock | 0.26 | **2.08** | **−2.64** | 0.46 | −0.74 |
| 16 | Toruń | −0.59 | 1.63 | −1.73 | −0.39 | **−2.64** |

**Table 11.** Results of the Mann test (*uF* statistics) for trends in precipitation characteristics in the seasonal and annual time scales; orange colour denotes positive *uF* values, blue colour negative. Numbers in bold indicate statistically significant changes at α level 0.05.

| No. | Station | Gini Index | | Maximum Dry Spell Length | Precipitation Total in September | Precipitation Total in October |
|---|---|---|---|---|---|---|
| | | Winter | Summer | | | |
| 1 | Skoczów | −0.08 | 0.61 | −1.91 | 1.85 | 1.17 |
| 2 | Bielsko-Biała | 0.98 | 1.78 | −1.46 | **1.98** | 1.05 |
| 3 | Katowice | **0.96** | −1.30 | −0.07 | 1.31 | 1.05 |
| 4 | Rycerka Górna | −0.47 | 0.38 | **−2.64** | 0.51 | 1.13 |
| 5 | Węglówka | **−2.09** | 1.31 | **−2.14** | 1.50 | 1.19 |
| 6 | Kraków | 0.54 | −0.48 | −1.07 | 1.28 | 1.14 |
| 7 | Kasprowy | **2.90** | −0.27 | −0.56 | **2.35** | 1.76 |
| 8 | Szaflary | **−2.40** | 0.10 | −1.14 | 1.62 | 1.58 |
| 9 | Białka Tatrzańska | −0.35 | −0.51 | −1.51 | 0.73 | 1.13 |
| 10 | Tarnów | −0.47 | 0.14 | −0.17 | 1.60 | 0.55 |
| 11 | Kielce | 1.62 | 0.23 | −0.74 | 0.60 | 1.16 |
| 12 | Lublin | −0.17 | −0.47 | −1.28 | **2.45** | 0.91 |
| 13 | Białystok | 1.49 | −0.25 | −1.04 | 0.43 | 0.28 |
| 14 | Pułtusk | 0.57 | −0.88 | −0.29 | 0.54 | 1.10 |
| 15 | Płock | 1.65 | 0.71 | 0.74 | −0.67 | 0.73 |
| 16 | Toruń | −0.43 | 1.31 | −1.26 | 1.07 | 0.47 |

**Table 12.** Results of the Mann test (*uF* statistics) for trends in precipitation structure in the winter season; orange colour denotes positive *uF* values, blue colour negative. Numbers in bold indicate statistically significant changes at α level 0.05.

| No. | Station | Classes of Daily Precipitation (mm) | | | | | | | |
|---|---|---|---|---|---|---|---|---|---|
| | | 0.0 | 0.1 | (0.1;0.5> | (0.5;1.0> | (1.0;5.0> | (5.0;10.0> | (10.0;20.0> | >20.0 |
| 1 | Skoczów | NA | **3.18** | 0.20 | 0.87 | 0.57 | −0.69 | −0.41 | 0.32 |
| 2 | Bielsko-Biała | 1.41 | **4.41** | −0.84 | −0.79 | −0.87 | −1.29 | −1.00 | 0.16 |
| 3 | Katowice | **4.12** | 0.50 | **−2.58** | −0.11 | −1.31 | −1.27 | 1.32 | 1.53 |
| 4 | Rycerka Górna | **4.74** | **5.03** | **4.12** | 0.06 | 0.19 | −1.29 | 0.34 | 1.02 |
| 5 | Węglówka | **2.60** | **4.49** | 1.74 | **4.46** | 1.74 | −1.29 | 0.93 | 0.08 |
| 6 | Kraków | 1.59 | 1.97 | 0.58 | −0.33 | 0.26 | −1.27 | 0.35 | 0.27 |
| 7 | Kasprowy | 0.03 | **2.22** | **2.19** | −0.24 | −1.87 | −1.39 | 0.92 | −0.75 |
| 8 | Szaflary | −1.52 | **2.30** | **2.11** | 0.77 | −0.04 | −1.29 | 0.60 | −1.20 |
| 9 | Białka Tatrzańska | **−4.83** | **2.50** | **3.32** | **3.22** | −0.83 | −1.39 | 0.14 | −0.07 |
| 10 | Tarnów | 1.17 | **2.78** | 0.86 | 0.48 | 0.23 | 0.64 | −0.94 | −0.15 |
| 11 | Kielce | **3.77** | 1.21 | 0.53 | 0.96 | **−2.01** | −1.35 | −1.11 | 0.59 |
| 12 | Lublin | **2.22** | 0.76 | −1.76 | −1.21 | 1.63 | −0.83 | −0.09 | 0.15 |
| 13 | Białystok | **1.96** | 1.31 | −1.14 | −0.31 | −0.96 | −0.68 | 0.51 | 1.36 |
| 14 | Pułtusk | −0.36 | **−3.80** | **−4.02** | −0.01 | −0.07 | −0.89 | **2.26** | 0.41 |
| 15 | Płock | **2.25** | **4.96** | 0.64 | 1.26 | −1.26 | −0.51 | 1.15 | 0.33 |
| 16 | Toruń | −1.65 | 1.85 | 0.41 | 1.76 | 0.19 | 0.24 | 1.01 | 1.67 |

**Table 13.** Results of the Mann test (*uF* statistics) for trends in precipitation structure in the summer season; orange colour denotes positive *uF* values, blue colour negative. Numbers in bold indicate statistically significant changes at α level 0.05.

| No. | Station | Classes of Daily Precipitation (mm) | | | | | | | |
|---|---|---|---|---|---|---|---|---|---|
| | | 0.0 | 0.1 | (0.1;0.5> | (0.5;1.0> | (1.0;5.0> | (5.0;10.0> | (10.0;20.0> | >20.0 |
| 1 | Skoczów | NA | **3.59** | **2.53** | **2.33** | 0.17 | −1.03 | 0.52 | −0.09 |
| 2 | Bielsko-Biała | −1.04 | **4.40** | 0.58 | 0.10 | −1.72 | 0.22 | 0.22 | 0.65 |
| 3 | Katowice | **2.08** | 1.84 | 1.58 | −0.76 | 0.41 | 0.65 | −0.84 | −0.18 |
| 4 | Rycerka Górna | **2.46** | **4.01** | **4.39** | **2.63** | 0.94 | 0.83 | −1.45 | 0.00 |
| 5 | Węglówka | 1.05 | **3.34** | **2.81** | **4.17** | 1.34 | −0.67 | 0.58 | −0.82 |
| 6 | Kraków | −1.04 | −0.25 | **−2.24** | **2.80** | −1.27 | 0.40 | **2.03** | −0.88 |
| 7 | Kasprowy | −1.89 | 0.65 | −0.88 | −1.49 | **−1.96** | −0.09 | 1.40 | 1.76 |
| 8 | Szaflary | −0.29 | 1.37 | 1.02 | 1.54 | −0.11 | 1.40 | −0.03 | 1.40 |
| 9 | Białka Tatrzańska | −1.37 | **2.03** | 1.76 | **4.77** | 0.66 | −1.00 | 0.37 | 0.50 |
| 10 | Tarnów | 0.13 | **3.42** | 0.96 | 1.54 | −1.43 | 0.52 | 0.40 | 1.66 |
| 11 | Kielce | **2.56** | **3.45** | 0.36 | 0.38 | 0.09 | −1.27 | 1.61 | 0.28 |
| 12 | Lublin | −0.63 | 0.74 | −1.17 | 0.48 | −0.27 | 0.32 | 1.90 | −0.27 |
| 13 | Białystok | −0.44 | **3.66** | −1.08 | 0.25 | −0.38 | −0.19 | 1.88 | 1.49 |
| 14 | Pułtusk | **−4.07** | **−2.08** | −0.15 | 0.02 | 0.42 | 0.12 | 1.89 | 0.56 |
| 15 | Płock | **3.37** | **3.73** | 0.29 | 1.24 | **−2.33** | 0.59 | 0.90 | −0.68 |
| 16 | Toruń | −0.79 | **2.65** | 1.46 | **2.05** | −1.48 | 0.68 | −0.19 | 1.13 |

## 4. Results

The results of the Mann test for the set of characteristics described in Section 3 are presented in Tables 5–13 in the form of the values of *uF* test statistics to compare the directions of changes represented by the sign of *uF*. The cells are coloured according to the significance of the results under the assumption of independence: light orange for an insignificant positive trend; dark orange for a significant positive trend; while light and dark blue were used for negative insignificant and significant trends at the level of 5%, respectively. The influence of a serial correlation is marked as described above in Section 4.

The colours in Table 5, representing the results of the Mann test (*uF* statistics) for trend in winter season daily flow characteristics along the Vistula show an almost consistent structure of changes with decreasing maxima and increasing minima, some being statistically significant. The number of days above the average maximum flow and below the average minimum flow decreases, which is a logical consequence of changes in the magnitudes of the maxima and minima. In most cases, the daily maxima occur earlier, but not significantly so. The centroid and the median time also reveal the same pattern along the Vistula course, which is insignificant but stable. It means that in the period 1951–2018, a non-significant shifting of runoff volume to the beginning of the winter season occurred. In 60% of stations the Gini index of concentration reveals a significant decrease in concentration (hence bigger uniformity) of flows. The assessments of changes in concentration by the moment of inertia test statistics are generally weaker than those obtained for the Gini index. Positive values of *uF* for inertia indicate a decreasing concentration, while negative values indicate an increasing concentration. The properties of these characteristics and their suitability for measuring the de-concentration of hydrographs ought to be tested in more detail.

The almost stable signs of trends along the Vistula course prove that large tributaries do not significantly change the flow characteristics of the recipient.

In the summer season (Table 6), only four values of the test statistic reached the significance level. All of them were found in two stations on the River Little Vistula, where human pressure in the form of mining water discharge is extremely important. Note that the *uF* for centroid and median are positive in all stations, which is contrary to the winter season. Surprising scores for minima show that there are no changes in low flows on the Vistula, which contradicts the common opinion of an intensification of hydrological droughts in the vegetation season. Almost stable signs of trends along the

Vistula course prove that big tributaries do not significantly change the flow characteristics in the recipient.

As may be expected, the results for the tributaries presented in Table 7 for winter and Table 8 for summer are more diverse than for the River Vistula due to a larger sensitivity to local conditions and pressures, although some similarities can also be found. These are: decreasing trends in the majority of stations in the magnitude of daily maxima (half of them significant) and the same for the duration, a large portion of minima show increasing trends with significant values in the basins in the San and the Middle Vistula and tributaries of the Narew basin, where a negative significant trend in the Gini index is observed. The centroid and the median with non-significant decrease are found for almost all stations.

There is no one specific pattern of trends in the summer season for the Vistula tributaries. Significant values (about 8% of all analysed cases) are spread over the range of stations and characteristics, which suggests that there are no visible effects of climate change or that the influence of local factors is stronger than the climatic changes that normally affect large areas.

Tables 9 and 10 present the results of the Mann test for trend in precipitation characteristics in the seasonal and annual time-scales while Table 11 presents the Gini index for summer and winter precipitation.

The analysis of the results does not allow us to draw clear conclusions. Significant trends in the number of days with precipitation are observed. No significant changes are found in the seasonal and annual precipitation totals except for in two stations with significant positive trends. The total precipitation amounts are rather stable but there are significant changes in opposite directions in the number of days with precipitation. The number of days with snow and snowfall totals generally decrease, but the changes are less accentuated, as may be expected. The share of snowfall in the annual precipitation totals and snow cover characteristics shows a strong decreasing trend, but not in all analysed stations. The maximum dry spell length decreases significantly in two stations in the mountainous part of the Vistula basin. The precipitation totals in September and October do not show significant changes except for three distant stations; however, all test statistics but one remain positive.

The changes in the precipitation structure presented for winter in Table 12 and for summer in Table 13 mainly affect the number of days with small precipitation up to 1 mm.

The results of linear trend assessments in the mean value and the standard deviation of instantaneous seasonal minima at the 15 stations along the Vistula course are presented in Table 14 and illustrated in Figure 6 for five selected stations on the four reaches of the Vistula, starting from Skoczów up to the closing station at Tczew.

As can be seen in Table 14, almost all trends in the mean values of instantaneous minima in the winter reveal a positive trend apart from four stations: Goczałkowice and Jawiszowice on the Little Vistula reach (negative trends) and Toruń and Tczew on the Lower Vistula reach (no visible trend). The trend in Goczałkowice and Jawiszowice differ from the adjacent stations, possibly because of water management on the Goczałkowice reservoir. Hydrological stations at Goczałkowice and Jawiszowice are situated just downstream and they reflect the water discharges from the reservoir. At Toruń and Tczew hydrological stations, the mean minimum flow is stabilised, showing practically no trend in the mean value. It is worthy to note that the Mann test results performed on daily minima data are comparable but not identical. The instantaneous minima differ slightly from the daily flow minima in smaller basins, and the greater the basin, the smaller the difference between them due to a greater inertia in the hydrological response. The standard deviations of winter minima differ slightly along the river course. Downward trends prevail in the standard deviation modelled by linear trends. Only at Bieruń Nowy, Szczucin and Puławy does the standard deviation increase over time where it is difficult to explain based on local conditions.

**Table 14.** Parameters of linear trends in the mean value and standard deviation of instantaneous seasonal minimal flows at the stations on the River Vistula; orange colour denotes positive trend, blue colour negative.

| No. | Station | Winter Minima | | | | Summer Minima | | | |
| --- | --- | --- | --- | --- | --- | --- | --- | --- | --- |
| | | Mean | | Standard Deviation | | Mean | | Standard Deviation | |
| | | Slope | Intercept | Slope | Intercept | Slope | Intercept | Slope | Intercept |
| 1 | Skoczów | 0.003 | 0.720 | −0.001 | 0.473 | 0.001 | 0.575 | 0.002 | 0.301 |
| 2 | Goczałkowice | −0.001 | 1.379 | −0.007 | 0.725 | 0.000 | 1.122 | −0.001 | 0.526 |
| 3 | Jawiszowice | −0.002 | 2.904 | −0.006 | 0.939 | −0.006 | 2.678 | −0.006 | 0.85 |
| 4 | Nowy Bieruń | 0.059 | 4.708 | 0.014 | 1.481 | 0.074 | 3.05 | 0.032 | 0.795 |
| 5 | Jagodniki | 0.087 | 50.095 | −0.019 | 12.139 | −0.129 | 51.343 | −0.105 | 13.94 |
| 6 | Szczucin | 0.349 | 77.855 | 0.011 | 22.539 | 0.311 | 78.547 | 0.172 | 16.063 |
| 7 | Sandomierz | 0.488 | 92.356 | −0.018 | 27.942 | 0.325 | 97.955 | 0.183 | 20.43 |
| 8 | Zawichost | 0.261 | 145.14 | −0.147 | 50.306 | 0.315 | 146.47 | 0.127 | 37.184 |
| 9 | Annopol | 0.338 | 150.87 | −0.005 | 48.181 | 0.233 | 155.153 | 0.181 | 38.038 |
| 10 | Puławy | 0.440 | 161.727 | 0.105 | 47.518 | 0.211 | 174.376 | 0.121 | 45.324 |
| 11 | Dęblin | 0.435 | 187.928 | −0.075 | 62.094 | 0.434 | 194.359 | 0.226 | 48.613 |
| 12 | Warszawa | 0.957 | 217.349 | −0.065 | 82.518 | 0.422 | 246.312 | 0.185 | 66.026 |
| 13 | Kępa Polska | 1.199 | 371.148 | −1.081 | 182.192 | 0.008 | 384.614 | 0.072 | 102.466 |
| 14 | Toruń | 0.141 | 400.539 | −0.711 | 130.009 | −0.325 | 399.589 | −0.316 | 123.544 |
| 15 | Tczew | 1.166 | 490.227 | −2.247 | 227.814 | 0.042 | 483.294 | −0.055 | 133.636 |

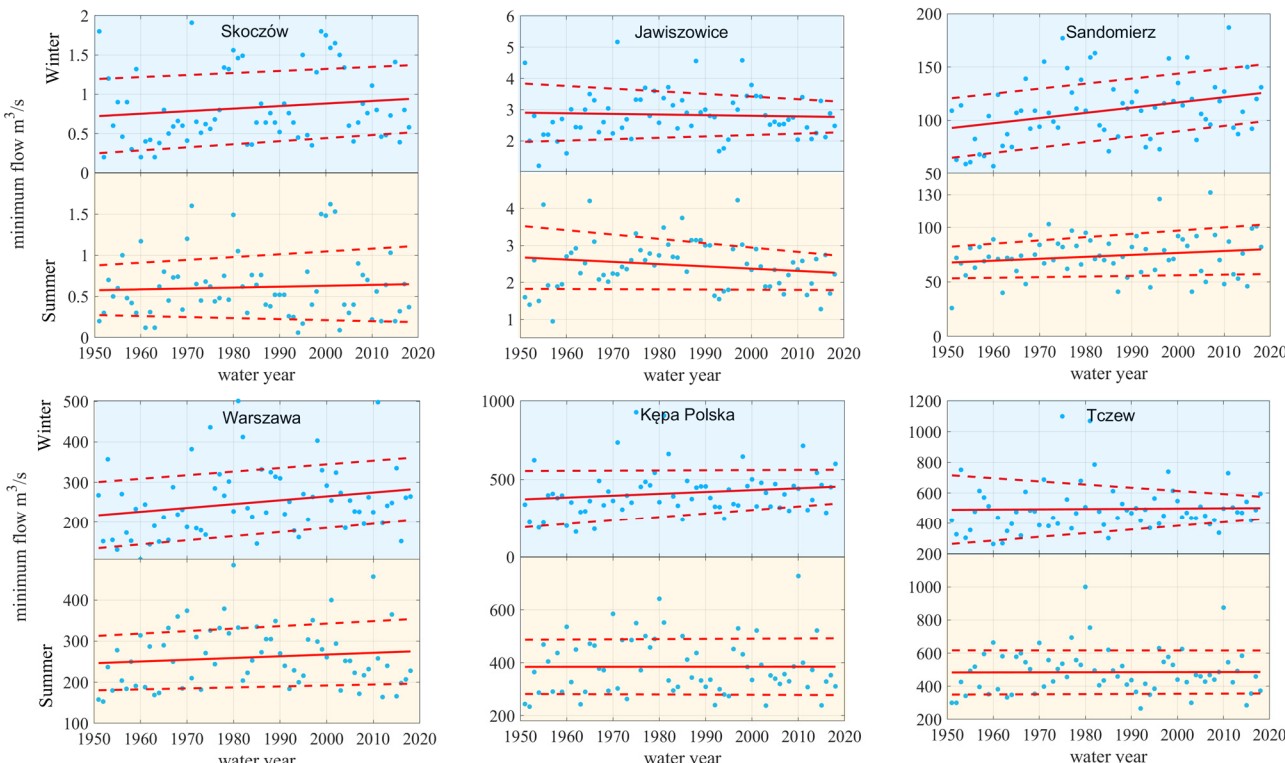

**Figure 6.** The linear trends in the mean and the standard deviation of seasonal instantaneous minima for selected stations along the Vistula.

In the summer, the Little Vistula reach reveals different trends in the mean value and the standard deviation. From almost stable values in Skoczów, decreasing trends in both mean and standard deviation are shown in Goczałkowice and Jawiszowice, while increasing trends are seen in Bieruń Nowy and decreasing in Jagodniki. Downstream from the Dunajec tributary, both trends become positive. On the Middle Vistula reach, i.e., from Zawichost to Warszawa stations, the mean value of summer minima slightly increase with

almost stable standard deviation in time. On the Lower Vistula, the mean value and the standard deviation remain practically constant.

One of the selected stations along the Vistula exhibits a different regime than in adjacent stations. This is the Bieruń Nowy station in the Little Vistula reach, which is highly influenced by mine water discharge [15]. In the period 1967–2013, on average 7.94 m$^3$/s of mine waters came from the drainage of hard coal mines in the Upper Silesia Coal Basin (USCB) was discharged into the River Vistula. The maximum discharge was reached in the period of largest extraction, i.e., the years 1979–1989, and especially 1985–1988 [15]. During this period, a mean discharge of mine water of 11.08 m$^3$/s was injected directly to the Vistula and to its tributaries in the Little Vistula and the Przemsza basins. The most affected rivers and stations were the River Przemsza with an average 6.66 m$^3$/s of mine water discharge and the Vistula at Jawiszowice and Bieruń Nowy stations. The greatest share of mine waters in the flow of the Little Vistula and the Przemsza basins ranged from 43% of the mean annual flow to 61% of the mean low annual flow at Czeladź on the River Brynica and the Vistula downstream from the Przemsza mouth (up to a 20% share in mean annual and mean low annual flow) to a 10% share at the Jawiszowice and Bieruń Nowy stations [37].

The same methodology of trend assessment was applied to Gini index series. The results are shown in Table 15 and Figure 7.

**Table 15.** Parameters of linear trends in the mean value and standard deviation of the seasonal Gini indices at the stations on the River Vistula; orange colour denotes positive trend, blue colour negative.

| No. | Station | Winter Minima | | | | Summer Minima | | | |
| --- | --- | --- | --- | --- | --- | --- | --- | --- | --- |
| | | Mean | | Standard Deviation | | Mean | | Standard Deviation | |
| | | Slope | Intercept | Slope | Intercept | Slope | Intercept | Slope | Intercept |
| 1 | Skoczów | −0.0002 | 0.4782 | −0.0002 | 0.0818 | 0.0010 | 0.5328 | −0.0002 | 0.0845 |
| 2 | Goczałkowice | −0.0008 | 0.4856 | −0.0002 | 0.0966 | −0.0003 | 0.5711 | 0.0002 | 0.0962 |
| 3 | Jawiszowice | −0.0010 | 0.4350 | −0.0006 | 0.1104 | −0.0011 | 0.5310 | 0.0009 | 0.0685 |
| 4 | Nowy Bieruń | −0.0013 | 0.3838 | −0.0004 | 0.094 | −0.0022 | 0.4892 | 0.0011 | 0.0709 |
| 5 | Jagodniki | −0.0012 | 0.3482 | −0.0003 | 0.088 | −0.0002 | 0.3428 | 0.0000 | 0.0885 |
| 6 | Szczucin | −0.0010 | 0.3378 | −0.0003 | 0.0927 | 0.0001 | 0.3248 | −0.0005 | 0.0918 |
| 7 | Sandomierz | −0.0012 | 0.3561 | −0.0003 | 0.0951 | 0.0003 | 0.319 | −0.0006 | 0.0923 |
| 8 | Zawichost | −0.0014 | 0.3725 | −0.0004 | 0.0995 | 0.0001 | 0.3183 | −0.0006 | 0.0915 |
| 9 | Annopol | −0.0012 | 0.3592 | −0.0002 | 0.0916 | 0.0002 | 0.3084 | −0.0004 | 0.0826 |
| 10 | Puławy | −0.0012 | 0.3426 | −0.0003 | 0.0972 | 0.0003 | 0.2866 | −0.0004 | 0.0813 |
| 11 | Dęblin | −0.0013 | 0.3404 | −0.0003 | 0.9048 | 0.0000 | 0.2863 | −0.0001 | 0.0750 |
| 12 | Warszawa | −0.0015 | 0.3262 | −0.0004 | 0.1007 | 0.0002 | 0.2469 | −0.0003 | 0.0785 |
| 13 | Kępa Polska | −0.0016 | 0.3196 | −0.0004 | 0.0896 | 0.0001 | 0.2319 | −0.0003 | 0.0674 |
| 14 | Toruń | −0.0016 | 0.3116 | −0.0004 | 0.0867 | 0.0001 | 0.2201 | −0.0003 | 0.0647 |
| 15 | Tczew | −0.0013 | 0.2847 | −0.0003 | 0.0792 | 0.0000 | 0.2059 | −0.0003 | 0.0590 |

In general, the Gini index decreases with the river's course, which is a consequence of the landform and alimentation system, but also the size of the basin and the tributary network structure. The average concentration of the Vistula winter daily flows ranges from about 0.47 in Skoczów to 0.24 in Tczew in winter and, respectively from 0.56 to 0.21 in summer.

In winter, all slopes of linear trend in the mean concentration measure remain negative and so are the slopes of the standard deviation linear trend. This indicates a greater uniformity of winter flows and a decreasing variability of the concentration from year to year. In summer the changes in the mean are more diversified, however only two of them are significant: positive at Skoczów and negative at Bieruń Nowy. From Jagodniki to Tczew the trend in the mean is small and nonsignificant with a decreasing standard deviation, which suggests some kind of stabilization of concentration at the end of the observation period.

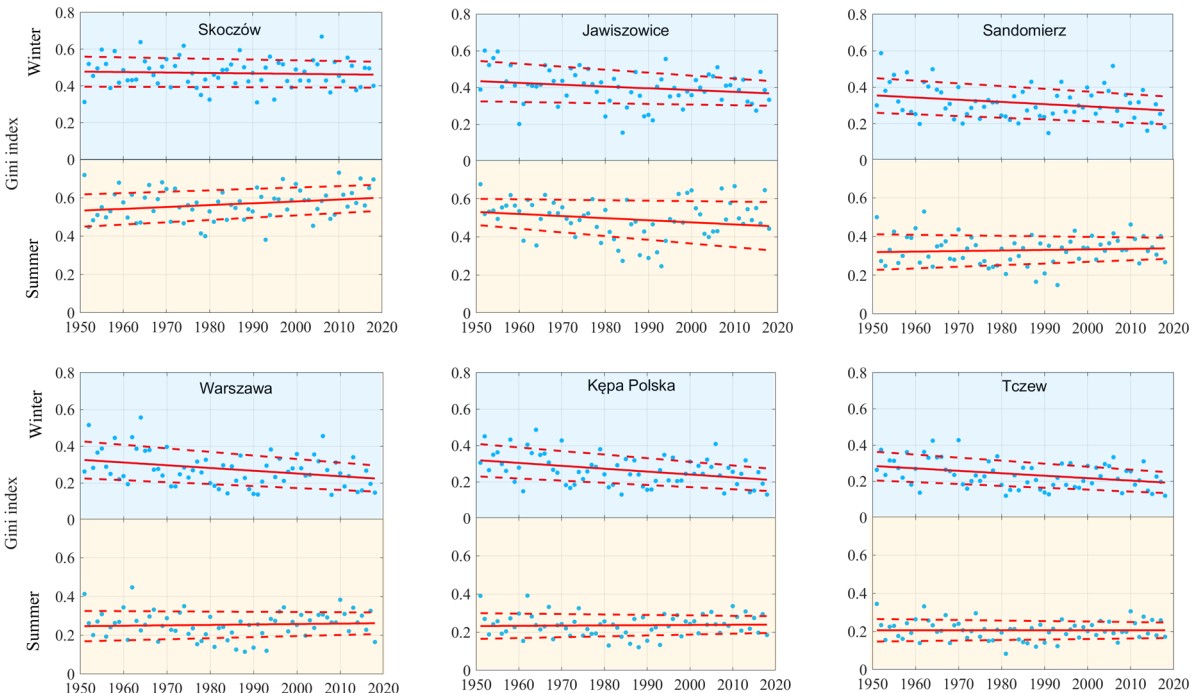

**Figure 7.** The linear trends in the mean and the standard deviation of the seasonal Gini index for selected stations along the Vistula.

## 5. Discussion

### 5.1. Runoff Characteristics

In the period 1951–2018, changes in the runoff characteristics in the hydrological stations along the course of the River Vistula were consistent or almost consistent in the winter season. It could be argued that a certain pattern of changes has been established. It seems to be in line with social perceptions and impressions, although not all changes have statistically significant trends. The runoff in the winter season has become more uniform with decreasing maxima and increasing minima of daily flows along with a stable mean runoff volume.

In the summer (Table 6), no important permanent changes in the runoff characteristics were found. However, attention should be paid to the insignificant trend in minimum daily flow, which contradicts the common impressions and media coverage of their significant decrease in recent years.

The results obtained for the Vistula tributaries show greater diversity with no regular patterns of changes. In winter, only one (almost) coherent area composed of the San, the tributaries on the Middle Vistula reach, and the tributaries of the Narew exhibit significant trends: positive in minimum daily flow and negative in the Gini index. In spite of differences between the results observed for the hydrological stations on tributaries, their integrated result has formed a pattern of the River Vistula.

### 5.2. Possible Reasons of Changes

In our opinion, there could be several reasons for the lack of significant trends in the summer daily minimum flows along the River Vistula.

The first is that the effects of global warming in terms of increased potential evapotranspiration were still too small in our region to cause significant and permanent changes in the daily minima. Research carried in the USA in southern Michigan in a similar climate zone to the majority of Polish territory (Dfb) showed that in a small natural catchment area with important changes in land use (type of vegetation cover) where the climate warmed by 1.14 °C, catchment-scale evapotranspiration had been stable over the study period of 50 years.



The second reason is due to the compensatory role of water reservoirs and water management in the basin and on the River Vistula itself consisting of regulation of flow during periods of drought. In addition, abstraction of water for agriculture is mainly taken from groundwater.

Additionally, the third, geomorphological reason, is that natural bottom erosion of the riverbed strongly accelerated by exploitation of sand and gravel has resulted in the river cutting into deeper horizons of groundwater and their drainage. River regulation, sediment retention in water reservoirs and bagging of navigable routes also play an important role here. The last reason can explain the contradiction between social perception of low flows and the results of statistical analysis. What one can observe when looking at the river is the water level not the volume of flow. The lowering of the river channel causes a decrease in the water level, especially visible at low levels, but the flow does not change, e.g., [4–6,38–40]. Perhaps the most spectacular example of such lowering is the situation in the Warsaw reach of the Vistula. During the low flow of 2015, the river unveiled monuments stolen during the war called the Swedish Deluge (1655–1660), which was transported by water to Sweden The loot sank in the Vistula and was recovered thanks to very low water levels in 2015. However, the flow in 2015 was the fourth lowest annual flow in the period 1951–2018. This means that if the river bottom were stable, the treasure would have been found much earlier in the former more severe drought period in 1921. In the period 1919–2015, the river's bottom in Warsaw had lowered by 225 cm [41]. On the Upper Vistula, this lowering is estimated to be approx. 300 cm [38]. In many places, difficulties in supplying water to the population and industry, socially perceived as a water shortage, are not the result of low river flows but of the difficulty of uptake (low water levels). Regularization of riverbeds resulting in shortening (by cutting meanders), narrowing (by levies and cross dikes construction) can accelerate and magnify the floods. As was shown in [38] in the upper Vistula basin, rainfall with a probability of 10–30% may now cause a 100-year flood on the Vistula. The impact of regularization on high waters is well recognized both in the world and in Poland, e.g., [38,42]. However, its impact on low flows still needs to be comprehensive researched.

### 5.3. Precipitation Characteristics

The changes in precipitation in the winter season show significant trends in snowfall and snow cover characteristics and the number of days with light precipitation. Since climatic change affects not only individual stations but large areas, the results show the important impact of local factors on some precipitation characteristics that can be stronger than the climatic drivers.

### 5.4. Comparison with Other Research Results

It is difficult to directly compare the results of this study with other research. This is mainly due to the different observation periods used, from paleo-climatic time spans to the last decade's data, and the scope of analysed characteristics. Climate variability can easily give rise to apparent trends when records are short—these are trends that would be expected to disappear once more data had been collected [43]. Researchers often use the observation period starting from 1971, which in some sources is considered representative of the assessment of Poland's surface water resources, allowing them to base their analysis of trends on it. The selection of the period is favourable from the point of view of the number of stations, which is greater than the number of stations with data since 1951. However, the data from 1971 do not cover the extremely dry period from around 1951–1964 [15,44]. Therefore, decreasing trends in low water characteristics should be expected while investigating this period. On the other hand, the choice of the period from 1951, starting with dry conditions, can lead to increasing trends in hydro-meteorological characteristics. Such trends were detected in this research. However, they prove that the later dry periods were less severe and could not change significant trends covering the entire analysed period.

Some comparisons, however, are possible. The results of the Gini index trend for daily precipitation obtained in this research overall conform with the results of [23], where no significant changes in concentration of daily precipitation have been found in seasons and annual values in the analysed period (1971–2010). The range of estimated indexes is also close.

In [44], the mean of minimum and maximum flows in two subsamples (1951–1970 and 1971–2005) of the analysed period 1951–2005 were compared, indicating that in most of the stations distributed over the Polish territory the mean maximum flow decreased and the mean minimum flow increased, mainly in the winter, which is in line with the results presented in Tables 5–8.

The direct comparison of the results of our trend studies in precipitation characteristics with the studies of climatologists is difficult due to different definitions of the year and hydrological and climatic seasons. Climatologists traditionally analyse precipitation patterns in calendar years and four three-month seasons shifted by one month backward from the beginning of the calendar year [45].

Wibig [46] concluded that the longest dry spells in precipitation for two stations located in the Vistula basin and one in the close vicinity of the watershed showed statistically significant downward trends so they became shorter. Another analysis in [47] pointed out that precipitation in Poland did not change greatly in the second half of the 20th century. Only the number of wet days (days with precipitation) has increased. Since the increase in the number of days with precipitation is not accompanied by an increase in the precipitation amount, the result is a decrease in the average precipitation on a wet day. These findings are consistent with the results obtained in [46–48] and in this research (Table 9).

The studies of meteorological droughts in Europe show that observational records from 1950 onwards and climate projections for the 21st century provide evidence that droughts are a recurrent climate feature in large parts of Europe, especially in the Mediterranean, but also in western, south-eastern and central Europe. Trends over the past 60 years show an increasing frequency, duration, and intensity of droughts in these regions, while a negative trend has been observed in north-eastern Europe [49]. In northern and eastern countries, drought severity shows a decrease linked to increased PET (excluding Romania) and precipitation. As both variables show significant trends (except in Poland and Slovenia), precipitation is the main driver in these regions [49].

### 5.5. General Remarks

Planners, designers, and other users of hydrological information today expect broad and accurate and certain information about the present and, above all, future characteristics of precipitation and runoff from hydrologists. However, at the moment we do not have any information other than that about past states and processes. Models using different versions of climate change scenarios can provide projections for the future, but there is no way to prove whether or not these results are. This can only be confirmed by future observations. Perhaps that is why constant monitoring and continuous supplementing and analysing of the sets of data is of great importance along with the development of the methods of these analyses.

### 6. Conclusions

- The hydrological regime of the River Vistula controlled and assessed at 15 gauging stations along its course forms a complex hybrid natural–human system, with insufficiently documented data and history of change. The observed runoff synthesizes this mosaic of processes and pressures. Hence, the accumulated tendencies can be read from different characteristics of runoff data. In contrast to a wide range of standardized indices such as SPI, SRI, and others that have become a common approach to analysis of riverine regimes in recent years, introducing some distortion of observation data, especially in the tails, the direct analyses of different runoff characteristics conserve their physical meaning and interpretation.

- The main tendencies found in this research is the increase of daily instantaneous minimum flow and growing uniformity of daily discharge in the winter season, significant in big part of observation series. Surprising scores for summer minima show that there is no important changes in low flows on the Vistula which denies the common opinions of an intensification of hydrological droughts in the vegetation season.
- Significant trends in snowfall and snow cover characteristics (the number of days with snow cover) were found, which is obviously the result of global warming.
- No significant trends in seasonal and annual precipitation totals and flow volumes were found. Changes in the seasonal precipitation structure revealed upward trends in the number of days with precipitation less than 1 mm in big parts of stations.
- The longest dry spell shows a weak decreasing tendency and the precipitation monthly totals in September and in October a weak increasing one. These tendencies can give the illusion of summer–autumn drought threat reduction if they continue. However, with increasing temperatures, which will result in an increase in field evaporation, the risk may increase significantly if the total rainfall remains unchanged.

**Author Contributions:** Conceptualization, E.B., R.J.R. and E.K.; methodology, E.B. and R.J.R.; formal analysis, E.B.; writing—original draft preparation, E.B.; writing—E.B., R.J.R. and E.K.; visualization, E.K. and E.B.; supervision, R.J.R.; project administration, R.J.R.; funding acquisition, R.J.R. All authors have read and agreed to the published version of the manuscript.

**Funding:** This research received no external funding.

**Institutional Review Board Statement:** Not applicable.

**Informed Consent Statement:** Not applicable.

**Data Availability Statement:** Data supporting reported results can be found in the Institute of Meteorology and Water Management.

**Acknowledgments:** This work was supported by the project HUMDROUGHT, carried out in the Institute of Geophysics at the Polish Academy of Sciences and funded by the National Science Centre (contract 2018/30/Q/ST10/00654). The hydro-meteorological data were provided by the Institute of Meteorology and Water Management (IMGW), Poland. The authors thank the Editor and anonymous Reviewers for their comments and suggestions that greatly improved the first version of the manuscript.

**Conflicts of Interest:** The authors declare no conflict of interest.

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
