# Peer review of "Temporal Changes in Flow Regime along the River Vistula"

_water, doi:10.3390/w13202840_

Round 1

Reviewer 1 Report

The Authors improved significantly the paper according to the reviewer requests.

Author Response

Thank you for your time spent on review our manuscript and your appreciation of our work.

Reviewer 2 Report

The authors have improved the content of the manuscript compared to its previous version. I have a few recommendations left.

  1. It is required to rewrite the Abstract since it should contain a laconic description of the results obtained (including quantitative characteristics).
  2. In the Keywords, the phrase "characteristics of seasonal and annual runoff and precipitation" should also be rewritten. Divide it into multiple keywords. The phrase "statistical analysis" is also inappropriate in Keywords. Statistical analysis is an inalienable part of modern research, including hydrological research. It is not a specific component of your study.
  3. Line 109: ecological flow? Environmental flow.
  4. Table 1. Please separate tenths with a period (dot), not a comma: 350.2, not 350,2. There are too many %-signs in the two rightmost columns. It is enough to indicate them only in the column headings.
  5. In all tables, there is no need for thousandths of values. Hundredths or tenths are enough.

Author Response

Answers to Reviewer #1

REV 1. The authors have improved the content of the manuscript compared to its previous version. I have a few recommendations left.

  1. It is required to rewrite the Abstract since it should contain a laconic description of the results obtained (including quantitative characteristics).

Answer: Yes, we have corrected the abstract according to your recommendation (lines 7-20). However, you should not expect a comprehensive description as we are limited to the required maximum 200 words of content. Now it is:

Abstract: The flow regime in the River Vistula is influenced by climatic and geographical factors and human intervention. In this study, we focus on an analysis of flow and precipitation variability in time and space following the course of the River Vistula. Multi-purpose statistical analyses of a number of runoff and precipitation characteristics were performed to present a general overview of the temporal and spatial changes. Since the important feature of the hydrological regime of Polish rivers is the seasonality of runoff associated with the occurrence of cold (winter) and warm (summer) seasons within a hydrological year, a seasonal approach is applied to describe specific seasonal features which can be masked when using annual data. In general, the results confirm popular impressions about changes in winter season runoff characteristics i.e., significantly decreasing daily maxima, increasing daily minima and decrease in concentration, and so a bigger uniformity of winter daily flows. An interesting behaviour of minimum flows in the summer season has been revealed, which is contrary to social perceptions and the alarming changes taking place in the other parts of the world. Also precipitation indexes related to the formation of droughts show no trends e.g., the mean value of the maximum dry spell length.

2. In the Keywords, the phrase "characteristics of seasonal and annual runoff and precipitation" should also be rewritten. Divide it into multiple keywords. The phrase "statistical analysis" is also inappropriate in Keywords. Statistical analysis is an inalienable part of modern research, including hydrological research. It is not a specific component of your study.

Answer: We have corrected the keywords according to your recommendations (line 21).

Now they are:

Keywords: Vistula basin; runoff, precipitation, climate change.

3. Line 109: ecological flow? Environmental flow.

Answer: Usually we use these terms interchangeably, but after analysing their exact definitions:

  • Ecological flows are considered within the context of the Water Frame Directive as “a hydrological regime consistent with the achievement of the environmental objectives of the WFD in natural surface water bodies”.
  • Environmental flows describe the quantity, timing, and quality of water flows required to sustain freshwater and estuarine ecosystems and the human livelihoods and well-being that depend on these ecosystems.

So, we agree with the Reviewer. There should be "environmental flow".

Corrected (line 105).

4. Table 1. Please separate tenths with a period (dot), not a comma: 350.2, not 350,2. There are too many %-signs in the two rightmost columns. It is enough to indicate them only in the column headings.

Answer: Thank you for noticing this error. Corrected (line 154).

5. In all tables, there is no need for thousandths of values. Hundredths or tenths are enough.

Answer: Tedious corrections due to the size of the tables and some difficulties to paste them into the text in the not deformed form, but happily corrected.

Thank you for your careful reading, remarks and corrections during all review procedure.

Best Regards,

Authors

Reviewer 3 Report

Reviewer’s Report on the manuscript entitled:

Temporal Changes in Flow Regime Along the River Vistula

The authors perform multi-purpose statistical analyses of runoff and precipitation characteristics to demonstrate a general overview of the temporal and spatial changes across the Vistula River Basin in Poland. The manuscript is well-written and suitable for publication in Water. However, I have several comments for the authors to improve their manuscript.

General Comments:

1) Please also include the following article in the Introduction:

https://doi.org/10.3390/w13030250

2) Neither the map in Figure 1 nor Tables 1 and 2 contains the latitude and longitude information of the stations. I suggest the authors at least put some dashed lines on the map showing the latitudes and longitudes.

3) Lines 239 and 251. Please add appropriate references.

4) Table 1. Do you mean daily-accumulated when you say precipitation total?

5) Table 2. Is it for one year or a decade or several decades? Please mention the total period in the caption.

6) Line 309. The trend analysis for streamflow time series is challenging. It depends on the sample size and whether the time series is equally sampled or not and the breakpoints/jumps and presence of the seasonal component in the time series. The following article shows that if the trend analysis using traditional regression analysis is applied to non-stationary time series (possibly with data gaps), then the trend can be estimated incorrectly (e.g., the true negative trend can be estimated as a positive trend). Furthermore, the more reliable trend estimations are those that consider the season-trend fit models which also consider the jumps or breakpoints: 

https://doi.org/10.1016/j.ejrh.2021.100847

Therefore, I also suggest the authors include the reference above in Lines 309 and 313 and also make a brief discussion in the Discussion section about the use of cross-wavelet analysis for estimating the coherency and phase differences between the precipitation and streamflow components as mentioned in the article above and also in the following article: https://doi.org/10.1016/j.ejrh.2015.11.003

7) The Discussion section (Line 489) needs to be organized perhaps by using subsections. Some of the texts toward the end of the Result section can also go to the Discussion section.

8) There are many tables showing Mann-Kendall test results (Tables 5-12). I suggest the authors consider moving some of them that are less important to an Appendix or Supplementary Materials. Furthermore, due to a formatting issue, I could not see half of Table 5.

Minor Editorial Comments:

I think in English it should be “Vistula River” not “River Vistula”. Please check and be consistent. In line 368, it is written correctly.

Line 8. Please insert a comma after “In this study”. Similarly, a comma after “In general” in line 18, a comma after “Therefore” in line 36, and similarly for lines 247, 267, 443, etc.

Line 36. Please replace “of crucial importance” with “crucial”.

Line 37. Please remove the year of publication “(2005)”. So, write it as Svensson et al. [7]. Same comment for lines 42, 44, etc. Please be consistent with the format and consult with the MDPI guidelines.

Lines 63-70. Grammar issue. Please instead of “questions” say “research objectives” because your question 2 in lines 66,67 is not written in the form of a question nor your question 3!

Line 82. Please remove “referring the reader to the above article”.

Line 151. Should be “Tables” not “Table”.

Please avoid using the roman numbers (XI, IV, V) in the tables. Instead mention the months in three letters like Jul, Aug, Sep, etc., and the years e.g., 2003, etc.  

Line 238. Please insert a dot after “in one point)”.

Thank you for your contribution

Best regards,

Author Response

Answers to Reviewer#3

REV 3

Reviewer’s Report on the manuscript entitled:

Temporal Changes in Flow Regime Along the River Vistula

The authors perform multi-purpose statistical analyses of runoff and precipitation characteristics to demonstrate a general overview of the temporal and spatial changes across the Vistula River Basin in Poland. The manuscript is well-written and suitable for publication in Water. However, I have several comments for the authors to improve their manuscript.

General Comments:

1) Please also include the following article in the Introduction:

https://doi.org/10.3390/w13030250

Answer: Yes, you are right that the Polish contribution to the research of seasons delineation is worth stressing. We added the citation and some comments in lines 128-132: Now it is:

The seasonal characteristics of flow differ, so their statistical properties ought to be analysed separately [15]. It is worth noting that an interesting method of hydrological seasons delineation in respect to river discharges and the base flow was presented recently in [21]. It is done by daily discharge transformation into three series reflecting three statistical features estimated for single-name days of a year from a multiyear: average value, variation coefficient, and autocorrelation.

2) Neither the map in Figure 1 nor Tables 1 and 2 contains the latitude and longitude information of the stations. I suggest the authors at least put some dashed lines on the map showing the latitudes and longitudes.

Answer: The tables almost exceed the limits of acceptable size, and it will be difficult to insert two additional columns without dividing the table into two, so we decided to change the maps.

Corrected (line 89).

3) Lines 239 and 251. Please add appropriate references.

Answer: Corrected. Three references have been added (lines: 219, 220, 251).

4) Table 1. Do you mean daily-accumulated when you say precipitation total?

Answer: We mean precipitation totals in winter, summer and year. Of course they are the sum of daily precipitation in the defined periods.

5) Table 2. Is it for one year or a decade or several decades? Please mention the total period in the caption.

Answer: Corrected (lines 159 -160). We added “ in the observation period (see Table 1)” in the caption of Table 2.

6) Line 309. The trend analysis for streamflow time series is challenging. It depends on the sample size and whether the time series is equally sampled or not and the breakpoints/jumps and presence of the seasonal component in the time series. The following article shows that if the trend analysis using traditional regression analysis is applied to non-stationary time series (possibly with data gaps), then the trend can be estimated incorrectly (e.g., the true negative trend can be estimated as a positive trend). Furthermore, the more reliable trend estimations are those that consider the season-trend fit models which also consider the jumps or breakpoints: 

https://doi.org/10.1016/j.ejrh.2021.100847

Therefore, I also suggest the authors include the reference above in Lines 309 and 313 and also make a brief discussion in the Discussion section about the use of cross-wavelet analysis for estimating the coherency and phase differences between the precipitation and streamflow components as mentioned in the article above and also in the following article: https://doi.org/10.1016/j.ejrh.2015.11.003

Answer: We have introduced the paragraph (lines 315-325) cited below in the section 3.2. Methods of analyses.

An important problem in hydrological studies, and especially in the realm of trend detection, is the number of measurements and observations and their uncertainty. Most of the methods of trend analysis require complete data in the analyzed period, but in practice such a situation is rather rare. Nowadays new methods are applied for trend detection e.g., the Least-Squares Spectral Analysis (LSSA) to analyse unequally spaced time series in the frequency domain. [38]. The method enables also the detection of jumps and breakpoints and can show how the components of the data change over time (see also [21]) and how climate change may impact the streamflow over time. It is expected that spectral and wavelet analyses can also help to provide more can help to provide more reliable climate and hydrological forecasts. It should be mentioned here that in the case of complete data the trend assessments are almost the same as obtained by traditional methods.

All the data from the hydrological and meteorological stations we have analysed have the complete daily and flow instantaneous extreme values but several daily values at Skoczów (about 0.1% of all values in this station).

7) The Discussion section (Line 489) needs to be organized perhaps by using subsections. Some of the texts toward the end of the Result section can also go to the Discussion section.

Answer: Subsections introduced (lines: 501, 520, 560, 567, 617). The last paragraph in “Conclusions” shifted to Section Discussion (lines 618 – 627).

8) There are many tables showing Mann-Kendall test results (Tables 5-12). I suggest the authors consider moving some of them that are less important to an Appendix or Supplementary Materials. Furthermore, due to a formatting issue, I could not see half of Table 5.

Answer: We discussed the way of presentation while conceptualising the research and we are aware that the tables are heavy and take much space in the manuscript body. However, it is difficult to present the results in more compact way without loss of important information. The presentation of the percent of stations with significant changes (applied in some papers) does not fulfil the role of the spatial visualisation of these changes. We considered the possibility of presentation on the maps but their scale and number would be greater than the volume of the tables.

It is also difficult to decide which tables are less important and can be presented in an appendix. Besides, in our opinion, the shifting of some tables to an appendix will damage the coherence of the presentation.

So, let it be.(?).

Minor Editorial Comments:

I think in English it should be “Vistula River” not “River Vistula”. Please check and be consistent. In line 368, it is written correctly.

Professor Ray Macdonald, a native speaker, who has read our manuscript, said that the correct version in English is "the River Vistula". Unfortunately, we have not corrected all expressions “the Vistula River” in the submitted text.

We attach here some explanations found on the Internet:

  1. After doing a little digging around, it seems that in Middle English the usual format was “river of X” - so the River Clyde, which flows through Glasgow in Scotland, would be “the River of Clyde”. In the Early Modern Period, the “of” disappeared, and it seems that in British English the original word order was kept, just without the missing preposition, while US usage seems to have analysed the expression a little differently and treated the name of the river as a modifier to the head noun “river” and changed the word order accordingly. British English, on the other hand, treats “River” almost as a honorific or title to be placed in front of the name.
  2. It is thought that this is a relic dating back to the Norman invasions and the influence the French had, as their practice would always be to put the word 'river' first.

There are many others like it. Seems it is correct to say the River Mississippi in UK English and the Mississippi River in US English. Since we formatted our manuscript in British English we ought to use the River Vistula consequently. We have corrected all appearances of the expression in the text (lines: 81, 84, 109, 135, 363, 377, 470, 473).

Line 8. Please insert a comma after “In this study”. Similarly, a comma after “In general” in line 18, a comma after “Therefore” in line 36, and similarly for lines 247, 267, 443, etc.

Answer: Corrected (lines: 35, 312, 486, 546).

Line 36. Please replace “of crucial importance” with “crucial”.

Answer: Corrected (line 35).

Line 37. Please remove the year of publication “(2005)”. So, write it as Svensson et al. [7]. Same comment for lines 42, 44, etc. Please be consistent with the format and consult with the MDPI guidelines.

Answer: Corrected (lines: 36, 41, 43).

Lines 63-70. Grammar issue. Please instead of “questions” say “research objectives” because your question 2 in lines 66,67 is not written in the form of a question nor your question 3!

Answer: Corrected (line 61).

Line 82. Please remove “referring the reader to the above article”.

Answer: Corrected (lines 78 - 79).

Line 151. Should be “Tables” not “Table”.

Answer: Corrected (line 150).

Please avoid using the roman numbers (XI, IV, V) in the tables. Instead mention the months in three letters like Jul, Aug, Sep, etc., and the years e.g., 2003, etc.  

Answer: Corrected in Table 1 (line 154).

Line 238. Please insert a dot after “in one point)”.

Answer: Done (line 238).

Thank you for your comprehensive remarks, suggestions and comments.

Best regards,

Authors

Round 2

Reviewer 3 Report

I would like to thank the authors for addressing my comments. The manuscript looks better now. Please when you proofread the article, please carefully check the references one by one to ensure they are correct and have a consistent format and style according to the MDPI guideline. For example, I found the following issues:

Line 740. should be "100847" not "1000847". Furthermore, the publication years must be in boldface (lines 685, 695, 713, 724, 728, 740).

Thank you for your contribution

Regards,

This manuscript is a resubmission of an earlier submission. The following is a list of the peer review reports and author responses from that submission.

Round 1

Reviewer 1 Report

Dear Authors, please find the comment in the attached file.

Reviewer 2 Report

  1. The discussion part is poorly presented in the manuscript. It is the weakest side of your study. In one way or another, your findings are widely observed in Central and Eastern Europe, but the authors do not discuss it. A small list of used literature tells me that the authors are not familiar with similar studies in neighboring countries. Moreover, in this section, the authors try to interpret somehow the results obtained, but they do not provide any evidence. For example, along lines 367-370. What are the rates of deep incision of the river to drain deeper aquifers in 50-70 years? Where is the quantitative evidence of this from geomorphological and hydrogeological studies? Quarrying in riverbeds can result in lower water levels downstream and, thus, create the illusion of a decrease in river flow. However, the authors again do not provide examples. Wasn't this kind of research done in Poland? Although your article gives preliminary results, it should be a complete scientific work, written like all other articles. Otherwise, I can regard it as a scientific report, but not as a scientific article.
  2. Authors should make the reader more aware of what droughts they are writing about in their work. What do you consider drought? What types of droughts do you analyze (meteorological, hydrological, or agricultural)?
  3. Lines 49-54. Why are you describing the structure of the manuscript if it is visible below? In these lines, you would better write more clearly about the objectives of your study compared to the previous works of other authors in the basin of this river. What is the step forward (novelty) in your research?
  4. The section "2. Study area and runoff and precipitation data" should be divided into two subsections. I recommend the section to be formatted as follows:

 2. Materials and Methods

  2.1. Study area

  2.2. Data and their sources

    2.2.1. Meteorological data

    2.2.2. Hydrological data

  2.3. Methods

    2.3.1. ...

    2.3.2. ...

    2.3.3. ...

The section "3. Characteristics of daily flow and precipitation processes" is methodological. Why do you consider it apart from the methodology?

  1. Table 4 and all next tables. Would you please write the period within which the trends were revealed?
  2. The section "6. Discussion and conclusions" should be divided into two separate sections.
  3. Table 1. The station at Kasprowy. Mean share of snowfall in the annual total - 61.9%; mean share of winter in the annual total - 40.7%. Is it right?